# GeoLoom: High-quality Geometric Diagram Generation from Textual Input

**XiaoJing Wei** [1]   **Ting Zhang** [1]   **Wei He** [2]   **Jingdong Wang** [2]   **Hua Huang** [1]

## Abstract

High-quality geometric diagram generation presents both a challenge and an opportunity: it demands strict spatial accuracy while offering well-defined constraints to guide generation. Inspired by recent advances in geometry problem solving that employ formal languages and symbolic solvers for enhanced correctness and interpretability, we propose GeoLoom, a novel framework for text-to-diagram generation in geometric domains. GeoLoom comprises two core components: an autoformalization module that translates natural language into a specifically designed generation-oriented formal language GeoLingua, and a coordinate solver that maps formal constraints to precise coordinates using the efficient Monte Carlo optimization. To support this framework, we introduce GeoNF, a dataset aligning natural language geometric descriptions with formal GeoLingua descriptions. We further propose a constraint-based evaluation metric that quantifies structural deviation, offering mathematically grounded supervision for iterative refinement. Empirical results demonstrate that GeoLoom significantly outperforms state-of-the-art baselines in structural fidelity, providing a principled foundation for interpretable and scalable diagram generation. The dataset is publicly available at: `github.com/BNU-ERC-ITEA/GeoNF`.

## 1. Introduction

Geometric diagrams play a crucial role in mathematical education, scientific research, and engineering design, serving as the core medium for concept visualization and knowledge transfer. Despite their significance, diagram generation remains predominantly manual and labor-intensive (Zhang et al., 2007; Yu et al., 2015). Conventional methods, primarily rule-based or interactive tools (Yu et al., 2015), depend heavily on user input and domain expertise, resulting in limited efficiency and scalability. These approaches typically require explicit specification of geometric primitives and constraints, rendering them inadequate for large-scale content creation. This highlights that automated models are urgently needed to convert natural language into accurate geometric diagrams.

In recent years, text-to-image generation models (Zhang et al., 2023; Jia et al., 2024), such as Stable Diffusion (Rombach et al., 2022), DALL·E (Ramesh et al., 2022), and their successors (Jiahui et al., 2022; Chang et al., 2023; Saharia et al., 2022; Ren et al., 2025; Mi et al., 2025; Kou et al., 2025), have achieved significant progress in synthesizing high-fidelity natural images from textual prompts. However, these models remain inadequate for generating geometric diagrams, primarily due to the sparse and highly structured nature of such visuals. Even with domain-specific fine-tuning, current models exhibit poor spatial accuracy, rendering the outputs unsuitable for educational contexts. This failure stems from a fundamental mismatch between the stochastic nature of generic diffusion models and the rigorous requirements of geometry, further exacerbated by the absence of high-quality paired datasets that align textual descriptions with precise diagrammatic data.

The core of the problem lies in the fact that geometric diagrams, unlike natural images, can not be evaluated based on subjective perceptual quality (Radford et al., 2021), geometric diagrams are inherently sparse, demanding exact placement of points, lines, and angles, and even a minor distortion undermines their mathematical validity (Chen et al., 2022). This divergence presents both a challenge and an opportunity. While current models struggle to capture precise spatial relationships and logical rules, the well-defined nature of geometry provides explicit, objective criteria. These criteria can serve as a principled foundation to calibrate generation quality, a potential that has yet to be fully exploited by existing pixel-based generative frameworks.

In this paper, we introduce GeoLingua, a novel generative-oriented formal language designed specifically for text-to-diagram synthesis in geometric domains. Central to GeoLingua is its role as an intermediate bridge that encodes geomet-

---

[1] School of Artificial Intelligence, BeiJing Normal University, China [2] BaiDu, China. Correspondence to: Ting Zhang <zhangting@bnu.edu.cn>, Hua Huang <huahuang@bnu.edu.cn>.

*Proceedings of the 43ʳᵈ International Conference on Machine Learning*, Seoul, South Korea. PMLR 306, 2026. Copyright 2026 by the author(s).

ric entities and relationships into a structured representation, facilitating the precise computation of coordinates necessary for high-fidelity rendering. This design draws inspiration from symbolic geometry solvers, where formal languages ensure logical consistency and computational rigor through unambiguous semantics (Chervonyi et al., 2025). Figure 1 shows this analogy. However, while conventional formal languages focus on problem-solving, they often lack the explicit constructive dependencies required for diagram synthesis. To bridge this gap, GeoLingua explicitly incorporates sequential construction orders and topological constraints, ensuring that every point and line is generated following a mathematically sound hierarchy. Leveraging this language, we further develop GeoNF, a large-scale paired dataset that aligns natural language descriptions with their corresponding GeoLingua representations, providing a robust benchmark for training and evaluating next-generation geometric generative models.

To this end, we propose GeoLoom, a two-stage framework that decouples logical reasoning from spatial realization through an autoformalization module and a geometric constraint solver. 1) The autoformalization module translates natural language descriptions into GeoLingua, a formal language specifically architected to capture the interplay between geometric primitives and their constructive dependencies. Within GeoLingua, geometric content is categorized into primitives and constraints. Primitives are organized into a hierarchy of free and dependent points, reflecting their causal construction order. Concurrently, constraints encode rigorous spatial relations, such as metric lengths and angular orientations, that dictate the underlying topology of the diagram. 2) The second stage transforms these formal specifications into a visual diagram by optimizing point coordinates. We employ a Monte Carlo-based optimization algorithm (Metropolis & Ulam, 1949) to efficiently navigate the high-dimensional solution space. This solver iteratively refines the positions of free points via stochastic perturbations, guided by a quantitative objective function that penalizes constraint violations. By minimizing this "geometric energy", the framework probabilistically converges on a precise spatial configuration that satisfies all mathematical prerequisites for accurate rendering.

Experimental results demonstrate that GeoLoom significantly outperforms state-of-the-art baselines across both qualitative and quantitative metrics, establishing a new benchmark for geometric accuracy and logical consistency. Furthermore, an analysis of inference latency reveals that GeoLoom achieves superior computational efficiency; this rapid generation capability underscores its suitability for real-time applications in educational technology and automated content creation. In summary, our primary contributions are as follows:

- **Formal Language and Dataset.** We introduce GeoLingua, a generative formal language that explicitly encodes the hierarchical structure of geometric primitives, constraints, and constructive dependencies. Building on this foundation, we curate GeoNF, a high-quality paired dataset aligning natural language with formal representations, which serves as a rigorous benchmark for geometric diagram autoformalization.

- **Methodological Framework.** We propose GeoLoom, a novel two-stage framework that decouples diagram generation into logical autoformalization and spatial realization. This framework integrates a Monte Carlo-based solver that iteratively optimizes point coordinates to satisfy complex geometric dependencies.

- **Empirical Validation.** We provide extensive experimental evidence demonstrating that GeoLoom achieves state-of-the-art performance in both accuracy and logical consistency. Furthermore, its superior computational efficiency validates its potential for real-time deployment in large-scale educational environments.

## 2. Related Work

**Text-to-Image Generation.** The field has progressed from GANs (Zhang et al., 2017; Kang et al., 2023) and autoregressive models (Chang et al., 2023) to diffusion-based approaches (Lian et al., 2024; Sun et al., 2024) capable of high-fidelity synthesis. However, these models often fail to maintain the rigorous structural integrity required for geometry. Existing diagram generation follows two primary paradigms: layout-guided, where LLMs plan spatial layouts for diffusion rendering (e.g., DiagrammerGPT (Zala et al., 2024)), and code-guided, which synthesizes programmatic code like TikZ from text (e.g., AutomaTikZ (Belouadi et al., 2024), DiagramAgent (Wei et al., 2024)). By contrast, formal systems like Penrose (Katherine et al., 2020) offer symbolic synthesis, yet they demand manual specification of complex logic, a limitation shared by dynamic geometry tools including GeoGebra and Sketchpad. To address these challenges, we introduce GeoLingua, a generative-oriented formal language that explicitly encodes geometric primitives and constructive dependencies. Integrated with a Monte Carlo-based solver, our GeoLoom framework enables automated synthesis of high-fidelity diagrams.

**Vector Graphics Generation.** Foundational work (Wu et al., 2025; Polaczek et al., 2025) in vector graphics synthesis includes DeepSVG (Carlier et al., 2020), which introduced a hierarchical generative model to disentangle high-level shapes from low-level Bézier and line commands. Building on this, SVGFusion (Xing et al., 2024) proposed a scalable text-to-SVG framework that combines a vector-pixel fusion VAE with a diffusion transformer to learn a

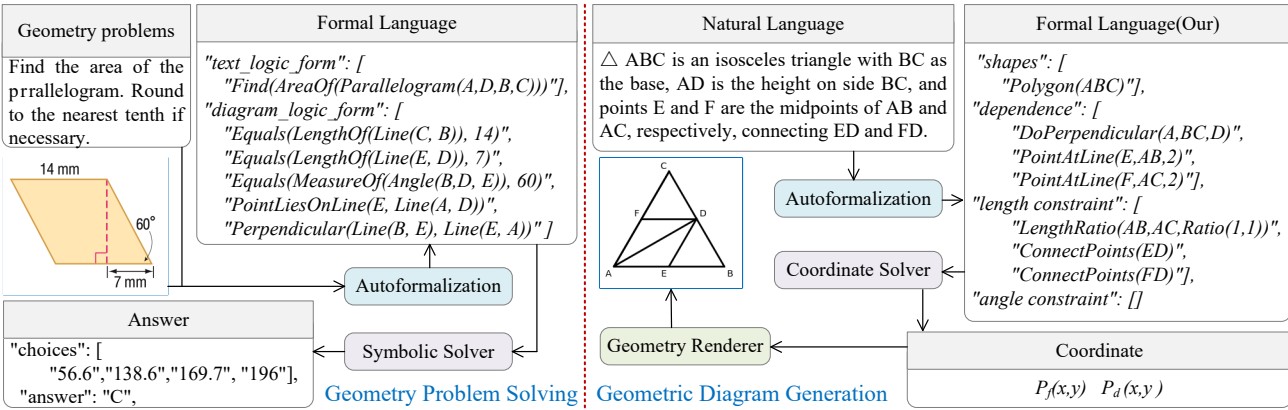

*Figure 1.* An overview of the analogy between geometry problem solving and geometric diagram generation, highlighting the central role of formal language in ensuring geometric correctness.

continuous latent space aligned with textual prompts. Despite these advancements in aesthetics, editability and scalability (Lopes et al., 2019; Carlier et al., 2020; Cai et al., 2023; Wu et al., 2023), most existing approaches lack explicit modeling of geometric constraints such as parallelism and perpendicularity, limiting their applicability in domains requiring strict geometric correctness.

## 3. GeoLingua

While formal languages have long supported geometry solving, existing representations often lack the capacity to capture topological dependencies essential for accurate diagram synthesis (Polu & Sutskever, 2020; Lu et al., 2021). This section introduces GeoLingua, a generation-oriented formal language designed for geometric constructions. To ensure its rigor, we conducted systematic consultations with mathematics experts. We first formally define the syntax and semantics of GeoLingua and then present GeoNF, a benchmark dataset to facilitate supervised training and evaluation.

**Syntax and Semantics.** We define four core components that underpin the generation of geometric diagrams: (1) Shape, (2) Dependence, (3) length constraint, and (4) Angle constraint. Building upon this abstraction, we introduce GeoLingua, a formal language framework designed to translate natural language (NL) problem descriptions into structured formal language (FL) representations. These representations are explicitly organized along the four axes mentioned above, enabling downstream coordinate optimization. The detailed definition can be found in Appendix A.1.

Shape: The foundational elements of a diagram are basic geometric primitives such as line segments, triangles, and circles. The defining vertices of these primitives are categorized as free points $P_f$, whose coordinates can be independently determined (e.g., the two endpoints $A$ and $B$ of the $AB$ line segment);

Dependence: Dependent points $P_d$ are determined based

on free points and geometric constraints (e.g., midpoints, perpendicular, intersections);

Length constraint: These include length ratios $C_{lin\_rat}$ and length relations $C_{lin\_rel}$, encoding proportional and relational constraints between line segments;

Angle constraint: These include fixed angle value $C_{ang\_val}$, angular ratios $C_{ang\_rat}$, and angular relations $C_{ang\_rel}$, specifying both absolute and relative angular properties.

**GeoNF Dataset.** Building upon the developed formal language framework, we present GeoNF, a curated dataset of geometry problems sourced from national educational examinations, specifically selected for their amenability to geometric diagram construction. To annotate the dataset, we recruited mathematics majors who were trained to translate natural language problem statements into structured formal representations. The detailed annotation process can be found in the AppendixA.2

GeoNF involves various types of geometric relations and geometric shapes, and we classify and statistically analyze them according to these relations and geometric shapes. The resulting GeoNF comprises 4,730 high-quality aligned pairs of natural and formal language representations, denoted as $\{N_n, F_n\}$, where $N_n$ represents the natural language description of a geometric problem, and $F_n$ denotes its corresponding formal expression. The specific distribution is shown in the Figure 2. The dataset is partitioned into 4,300 training pairs and 430 test pairs, enabling models to learn mappings from informal language to formal specifications, thereby supporting constraint understanding.

## 4. GeoLoom

Based on GeoLingua, we propose GeoLoom, a two-stage framework for high-precision text-to-diagram generation in geometry domains. GeoLoom bridges natural language descriptions and structured geometric diagrams by integrating

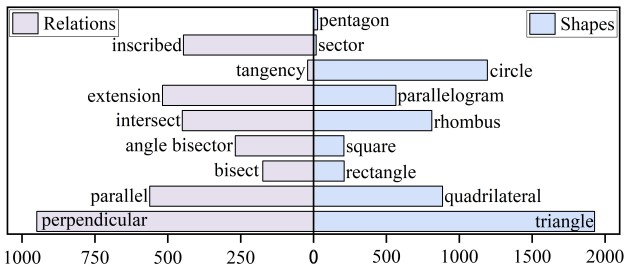

*Figure 2.* Geometric relations and shapes distributions in GeoNF.

an autoformalization module with a coordinate solver. An overview is illustrated in Figure 3.

### 4.1. Autoformalization

We investigate two complementary approaches: (1) a training-free, prompt-based method, and (2) a fine-tuning method. In both settings, the generated formal expressions are subsequently processed via Monte Carlo optimization to compute precise coordinates, further enabling high-quality generation.

**Training-free.** We develop a prompt-engineering pipeline that interfaces directly with LLMs (specifically DeepSeek-V3 (Liu et al., 2024)), enabling translation from natural language to structured formal representations without additional training. This pipeline integrates a validation mechanism to ensure syntactic and semantic compliance with the target formal language. Using the GeoNF test set, the process unfolds in three stages: (i) prompt construction guided by geometric semantics, (ii) structured generation of candidate formal expressions, and (iii) validation-based filtering (prompt and detailed validation methods can be found in Appendix B.1 and Appendix B.2 respectively).

**Fine-tuning.** To enhance model adaptability for formal language generation in geometric contexts, we conduct fine-tuning experiments across different model families (LLaMA (Grattafiori et al., 2024) and Qwen (Bai et al., 2023)) with various parameter scales. Each model is fine-tuned on the training split of the GeoNF dataset, which contains aligned pairs of natural language inputs and corresponding formal representations.

This dual-path investigation offers a comprehensive comparison of zero-shot and training-based strategies for autoformalization, establishing a foundation for robust formal language generation.

### 4.2. Coordinate Solver

**Geometric Constraint Deviation.** The well-structured nature of geometric diagrams enables the use of explicit, quantifiable evaluation metrics, thereby providing a principled basis for supervising the diagram generation process. While

general image evaluation (Lin et al., 2024) typically focuses on the presence of basic visual features and semantic alignment with textual input (Radford et al., 2021), the evaluation of geometric diagrams necessitates a stricter adherence to mathematical precision. Specifically, this includes the accurate placement of point coordinates and compliance with geometric constraints related to line segments and angles. We hence identify five fundamental types of geometric constraints that govern the structural fidelity of diagrams and formalize each as follows (In the below equation, [] is an Iverson bracket (Knuth, 1992), always 0 or 1.):

- **Length ratio.** Given a ground-truth length ratio $L_i^{gt}/L_j^{gt} = R_{gt\_line}$, the length ratio for generated diagram is $L_i^{gen}/L_j^{gen} = R_{gen\_line}$. The evaluation metric for this constraint is defined as:

$$\mathcal{C}_{lin\_rat} = R_{gen\_line}/R_{gt\_line}. \tag{1}$$

- **Length relation.** For a relational expression between two line segments $L_{left} \odot L_{right}$, and $\odot \in \{=, >, \geq, <, \leq\}$, the evaluation metric is given by:

$$\begin{aligned} \mathcal{C}_{lin\_rel} = [\odot = \text{``} = \text{''}] \cdot L_{left}/L_{right} \\ + [\odot = \text{``} \star \text{''} \text{ and } L_{left} \star L_{right}], \end{aligned} \tag{2}$$

where $\star$ can be $> (\geq)$ or $< (\leq)$, $L_{left}$ ($L_{right}$) represents the left-hand (right-hand) side of the length relational expression.

- **Angle value.** Given a ground-truth angle $\theta_{gt}$ and a generated angle $\theta_{gen}$, the constraint evaluation is:

$$\begin{aligned} \mathcal{C}_{ang\_val} = [\theta_{gt} \neq 0] \cdot \frac{\theta_{gen}}{\theta_{gt}} \\ + [\theta_{gt} = 0 \text{ and } \theta_{gen} = 0]. \end{aligned} \tag{3}$$

- **Angle ratio.** Given a ground-truth angle ratio $\theta_i^{gt}/\theta_j^{gt} = R_{gt\_angle}$, and a generated ratio $\theta_i^{gen}/\theta_j^{gen} = R_{gen\_angle}$, the constraint metric is:

$$\mathcal{C}_{ang\_rat} = R_{gen\_angle}/R_{gt\_angle}. \tag{4}$$

- **Angle relation.** For an angle relation $A_{left} \odot A_{right}$, with $\odot \in \{=, >, \geq, <, \leq\}$, the evaluation is:

$$\begin{aligned} \mathcal{C}_{ang\_rel} = [\odot = \text{``} = \text{''}] \cdot A_{left}/A_{right} \\ + [\odot = \text{``} \star \text{''} \text{ and } A_{left} \star A_{right}]. \end{aligned} \tag{5}$$

where $\star$ can be $> (\geq)$ or $< (\leq)$, $A_{left}$ ($A_{right}$) are the left-hand (right-hand) side of the angle relation expression.

The five values above constitute the normalized constraint satisfaction scores in the interval $[0, 1]$, where 1 indicates

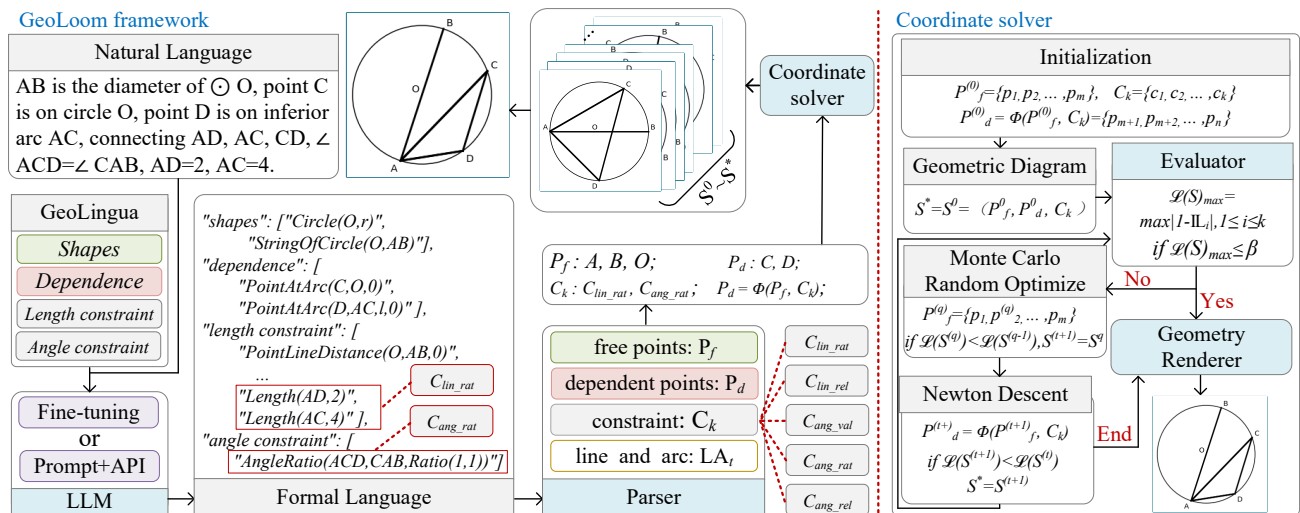

*Figure 3.* Illustrating GeoLoom framework: an autoformalization module and a coordinate solver.

perfect satisfaction of the corresponding constraint, and lower values reflect increasing degrees of deviation.

**Monte Carlo-based Algorithm.** To compute precise point coordinates in geometric diagrams, we adopt a Monte Carlo-guided coordinate solver, which iteratively perturbs the coordinates of free points, propagates updates to dependent points via constraint functions, and minimizes geometric constraint violations. This approach effectively balances local optimization and global search, thereby avoiding entrapment in suboptimal configurations.

Formally, from the formalized language, we regard a geometric diagram as $S = (P_f, P_d, \mathcal{C})$. $P_f = \{p_1, \ldots, p_m\}$ denotes the set of free points whose coordinates $\{(x_1, y_1), \ldots, (x_m, y_m)\}$ can be independently adjusted. $P_d = \{p_{m+1}, \ldots, p_n\}$ denotes the set of dependent points, which are computed from the geometric constraints and the positions of $P_f$.

For evaluation, we collect all scalar constraint scores into a single set $C$, which equals

$$C_{lin\_rat} \cup C_{lin\_rel} \cup C_{ang\_val} \cup C_{ang\_rat} \cup C_{ang\_rel}, \quad (6)$$

where each element $C_k \in C$ is the value of one constraint metric defined in the five families above. The optimization objective is to maximize the overall satisfaction of all constraints. We define the deviation loss as,

$$\mathcal{L}(S) = \max_{C_k \in C} |1 - C_k|. \quad (7)$$

Since each $C_k \in [0, 1]$, minimizing $\mathcal{L}(S)$ is equivalent to maximizing the minimum constraint satisfaction score. We iterative point coordinates until the deviation falls below the tolerance threshold $\alpha$. The full procedure is detailed in Appendix C.1.

**Geometric Diagram Render.** To support high-quality visualization and compatibility with computational geometry tools, we use Matplotlib for rendering geometric diagrams. The renderer parses the formal language to extract points $P$ and line segments $L$, then applies a dynamic coordinate adjustment mechanism that computes appropriate scaling and translation based on the spatial distribution of points, which prevents overflow. Implementation details are provided in Appendix C.2.

# 5. Experiments

**Implementation Details.** The coordinate solver involves three principal hyperparameters: the loss convergence threshold $\alpha$, the number of inner loop iterations $Q$, and the number of outer loop iterations $T$. Specifically, we set $\alpha = 0.05$, $Q = 1000$, and $T = 1000$. Complete implementation details, including prompt formats and training configurations, are provided in the Appendix B.

## 5.1. Results

We conduct a comprehensive evaluation of GeoLoom by benchmarking it against strong baselines. Additionally, we analyze the computational efficiency of GeoLoom, highlighting its scalability and inference speed.

**Qualitative Evaluation.** Figure 4 presents representative visual examples comparing our method with three baselines: the state-of-the-art text-to-image model Seedream3.0 (Gao et al., 2025), Seedream5.0 and the domain-specific text-to-diagram generator AutomaTikZ (Belouadi et al., 2024). To verify the effectiveness of GeoLingua, we incorporate GeoLingua knowledge prior into the baseline methods. As shown, our approach, either training-free or fine-tuned, consistently generates diagrams that rigorously satisfy geometric constraints. In contrast, the baseline methods, though capable of producing diagram-like visuals, frequently violate these constraints. This superior performance arises from

| Ground Truth | | Ours | | Without Prior Knowledge | | | With Prior Knowledge | | |
|---|---|---|---|---|---|---|---|---|---|
| Natural Language | Origin-img | Training-free | Fine-tuned | AutomaTikZ | Seedream3.0 | Seedream5.0 | AutomaTiKZ | Seedream3.0 | Seedream5.0 |
| The triangle ABC is an inscribed equilateral triangle of circle O, point O is the center, po-ints D and E are on sides AC and AB respec-tively, connecting OD and OE, DA=BE. | | | | | | | | | |
| In triangle ABC, points D, E, and F are res-pectively the midpoints of AB, AC, and BC, connecting DE, EF, and DF. Points P, M, and N are respectively the midpoints of DE, DF, and EF, con-necting PM, PN, and MN. | | | | | | | | | |
| I n the four sides ABCD, ∠BAD=120∘, ∠ABC=∠ADC=90∘, find points M and N on BC and CD respectively, and connect AM and AN. | | | | | | | | | |
| In rectangle ABCD, point E is on edge AB, connected to DE, which is the bisector of ∠ADC. Point F is on the extension of DE, connected to BF, ∠BFE = 90∘, connected to AF, CF, CF, and AB intersecting with G. | | | | | | | | | |

*Figure 4.* Qualitative comparison with AutomaTikz (based on LLaMa7b), Seedream3.0 and Seedream5.0. With Prior Knowledge means that the model incorporates GeoLingua. (The introduction of baselines can be found in the Appendix D.1) and the corresponding formal language and the more results can be found in Appendix D.3

*Table 1.* Quantitative comparison. Manual represents the accuracy of manual examination. Average represents the average value of LCI and ADI. The training-free model is DeepSeek-V3.

| Model | Manual (Unit:%) | LCI↓ | ADI↓ | Average↓ |
|---|---|---|---|---|
| *Fine-tuned* | | | | |
| LLaMa3.1-8b | 79.06 | 0.996 | 2.104 | 1.550 |
| LLaMa3.2-3b | 77.21 | 0.951 | 2.104 | 1.549 |
| LLaMa3-8b | 75.58 | 1.326 | 2.666 | 1.996 |
| Qwen2.5-7b | **85.34** | 0.747 | 1.818 | **1.283** |
| Qwen2.5-14b | 83.72 | 0.871 | 1.083 | 1.427 |
| *Training-free* | | | | |
| DeepSeek-V3 | 81.16 | 0.926 | 1.747 | 1.337 |

our integration of formal representations and a constraint-aware coordinate solver, ensuring both semantic alignment and geometric validity.

We further evaluate our method on more complex IMO-style datasets and compare with established tools that require human intervention, namely Penrose (Katherine et al., 2020) and GeoGebra (Hohenwarter & Preiner, 2007). As illustrated in Figure 5, while all three methods achieve mathematical correctness, they differ significantly in efficiency and accessibility. GeoLoom automates the entire synthesis process in approximately 60 seconds per diagram. In contrast, Penrose imposes a high cognitive load, requiring users to learn and manually author complex symbolic scripts, while GeoGebra necessitates manual construction, averaging 300 seconds of expert labor per diagram. Additionally, we conducted experiments using DeepSeek-V3 to directly convert text into geometric schematic diagrams via SVG (Direct-to-SVG). Experiments show that although SVG is a drawing language, it requires precise geometric point coordinates as a prerequisite. Therefore, models cannot directly use SVG to draw correct geometric diagrams.

**Quantitative Evaluation.** To complement the qualitative

analysis, we quantified the error values between the generated diagrams and the target diagrams. We leverage our geometric constraint deviation score and group them into two metrics that can be used as quantitative indicators. The first is Line Consistency Index (LCI) capturing deviations related to line segments,

$$LCI = 1 - \frac{\sum \left( \mathcal{C}_{lin\_rat} + \mathcal{C}_{lin\_rel} \right)}{N_l}, \qquad (8)$$

where $\mathcal{C}_{lin\_rat}$ and $\mathcal{C}_{lin\_rel}$ represent the length ratio and length relation constraint scores, respectively. $N_l$ represents the number of all elements in the length constraint set. The second is Angle Deviation Index (ADI) capturing deviations related to angular properties,

$$ADI = 1 - \frac{\sum \left( C_{ang\_val} + C_{ang\_rat} + C_{ang\_rel} \right)}{N_a}, \qquad (9)$$

where $\mathcal{C}_{ang\_val}$, $\mathcal{C}_{ang\_rat}$ and $\mathcal{C}_{ang\_rel}$ represent angle value, angle ratio, and angle relationship constraints, respectively. $N_a$ represents the number of all elements in the Angle constraint set.

We report the quantitative results by conducting a comparative analysis of different LLMs under the finetuning setting, and the results are shown in Table 1. To ensure the reliability of our proposed quantitative indicators, we perform a human-centric validation where a manual evaluation team reviewed the generated diagrams for mathematical accuracy. Our evaluation includes two methodological paradigms for formal language generation: supervised fine-tuning and training-free (zero-shot) inference, both followed by our coordinate solver for diagram rendering. As shown in Table 1, the Qwen2.5-7B model, when fine-tuned with our formal language, achieves the highest generation accuracy. This highlights the efficacy of incorporating formal language

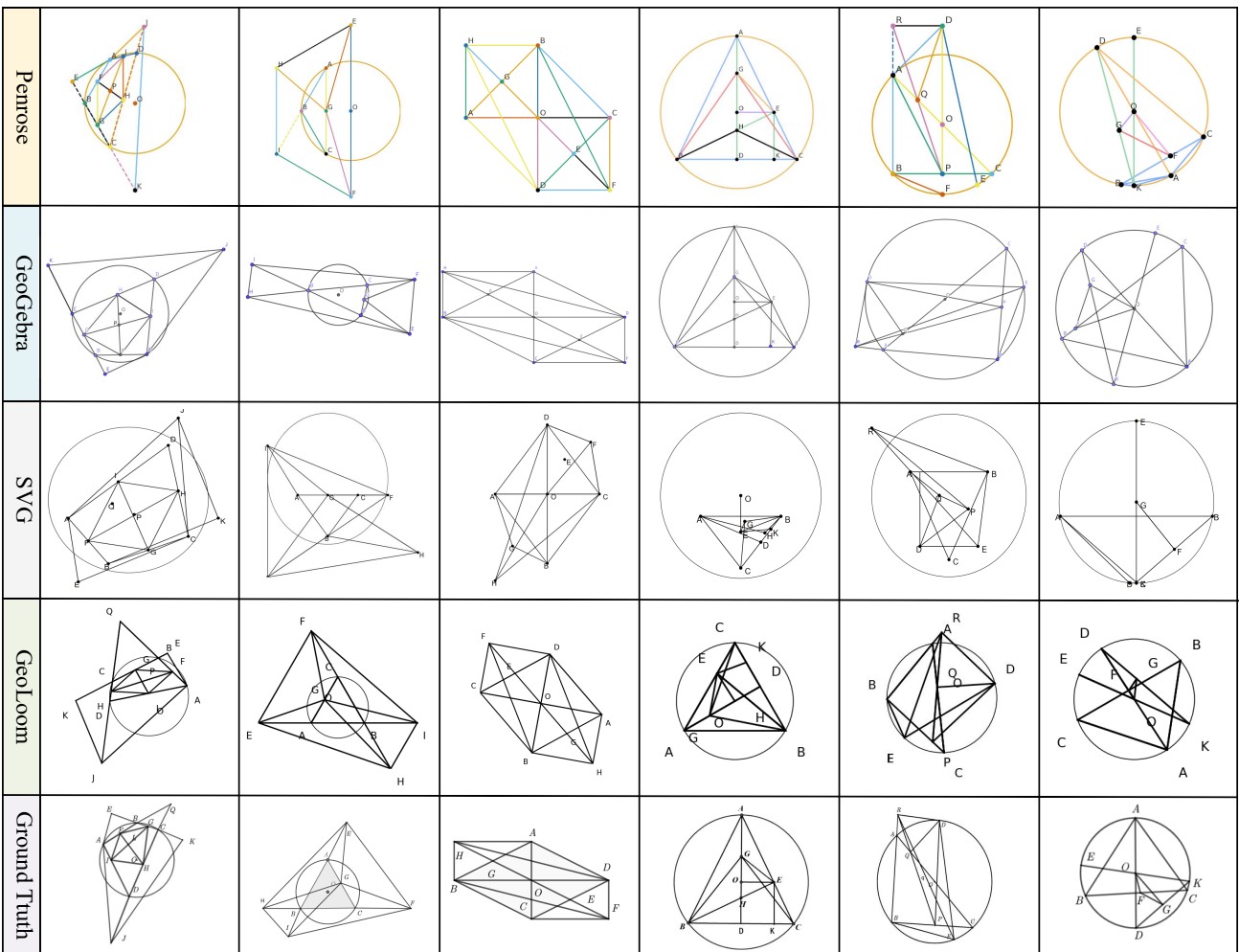

*Figure 5.* Complex IMO-style diagram generation comparison with Penrose, GeoGebra, and Direct-to-SVG. The results of GeoLoom from training-free mothod with DeepSeek-V3. Direct-to-SVG experiment using DeepSeek-V3. Natural language inputs appear in Figure 11.

supervision to calibrate the model's spatial and logical reasoning during the fine-tuning process. We prioritized LCI and ADI to evaluate diagrams, because general images metrics(e.g., CLIP (Radford et al., 2021)) are unsuitable for diagrams. The limitations of general indicators in the diagrams can be analyzed in detail in the AppendixD.2

**User Study.** To evaluate the effectiveness from a human-centric perspective, we conduct a user study on a randomly selected sample of 100 geometric diagrams. Ten participants with backgrounds in mathematics / computer science were recruited to evaluate each diagram along two dimensions: (i)(Quality) image quality, namely visual plausibility; and (ii)(Alignment) textual alignment, the consistency between the diagram and textual description. For each instance, participants were shown diagrams generated by four different methods (including ours) in the randomized order, accompanied by the original textual description. The fifth option, "None of the above," was provided to allow participants

to reject all diagrams if none were satisfactory. Table 2 presents the aggregated results. Our method consistently outperforms all baselines across both evaluation criteria. A small proportion of samples were rated unsatisfactory for all methods, suggesting that evaluators exercised critical judgment rather than responding arbitrarily.

*Table 2.* User study in image quality and textual alignment.

| Model and Method | Quality | Alignment |
|---|---|---|
| AutomaTikZ | 0 | 0 |
| Seedream3.0 | 0 | 0.2 |
| Ours (training-free) | 39.2 | 40.5 |
| Ours (fine-tuned) | 55.1 | 53.9 |
| None of the above | 5.7 | 5.4 |

**Efficiency Results.**

To evaluate the efficiency of the Monte Carlo method, we conducted experiments and measured the generation time

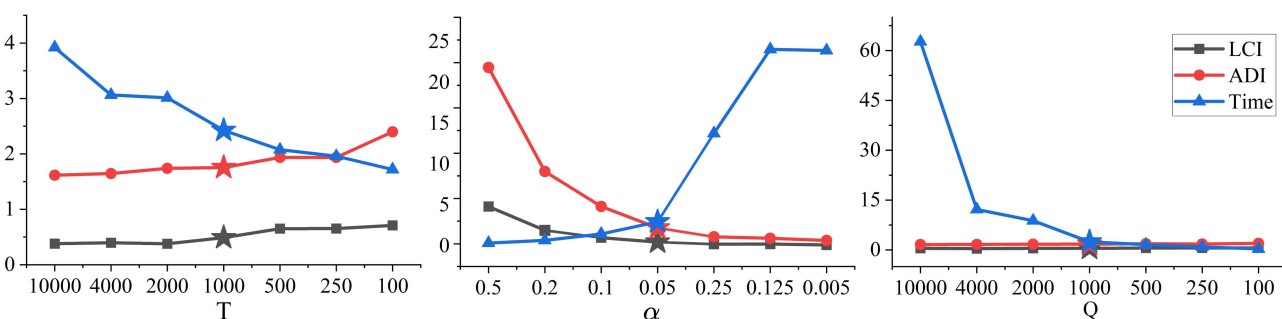

*Figure 6.* Parameter analysis. The changes of LCI,ADI and Time with the increase of x-axis multiplier for $\alpha$, $T$ and $Q$ .

of each instance. The results can be found in the Table 3. Nearly 70% of the generated images were generated within 10 seconds, demonstrating the practical effectiveness of this method. The efficiency of the untrained form on the training set matches well with the time efficiency on the test set. This demonstrates the stability of the Monte Carlo method and emphasizes its reliability in time sensitive applications. The experimental configuration can be found in the Appendix D.4

### 5.2. Analysis

**Failure Case Analysis.** We analyze failure cases of the Qwen2.5-7B experiments from both qualitative and quantitative perspectives. A sample is regarded as a failure if either the formalization or the generated diagram is incorrect. Most formalization errors arise from irregular or ambiguous natural-language descriptions that lead to topological mistakes. Even with correct formal inputs, the Monte Carlo optimization may still fall into local minima or produce overlapping configurations. Local minima constitute the primary failure mode, accounting for about 8% of the test set. More examples and statistics are provided in Appendix E.1.

**The Effect of Validation Filters.** We here ablate the impact of validation filters on automatic formalization. As shown in Table 4, incorporating these filters into training-free autoformalization substantially improves the quality of the generated geometric diagrams, while markedly reducing both LCI and ADI scores. Notably, the filtered autoformalization results approach the quality of manually annotated formal languages. These findings demonstrate that validation filters effectively correct non-standard or noisy constraints in the autoformalization process, thereby significantly enhancing the overall reliability and expressiveness of the resulting formal language.

**The Effect of Different Geometric Constraint Deviations.** To evaluate the impact of different geometric constraints on the final generation quality, we further conduct constraint ablation experiments. As detailed in Table 4, isolating length constraints yields the highest length accuracy (indicated by the lowest LCI) and the most rapid convergence. This suggests that length-based objectives are computationally robust and relatively straightforward to satisfy within the optimization space. In contrast, preserving only angle constraints results in comparable angular precision, indicating that angular relations are more sensitive to stochastic perturbations. Crucially, while single-constraint models may excel in their specific metrics, they fail to maintain the overall topological integrity of the diagram.

**Parameter Analysis.** We investigate the impact of three key parameters, loss convergence threshold ($\alpha = 0.05$), inner-loop iterations ($Q = 1000$), and outer-loop iterations ($T = 1000$), on the efficiency and quality of geometric diagram generation. Figure 6 summarizes generation time and diagram accuracy under six varying configurations(x-axis). The convergence threshold $\alpha$ defines the stopping criterion for optimization. Smaller values enforce stricter convergence, yielding higher-precision diagrams at the cost of increased computation, while larger values accelerate termination but may compromise spatial accuracy. The outer-loop count $T$ governs global exploration, enhancing robustness by escaping local optima. The inner-loop iteration count $Q$ controls local refinement granularity. Higher $T$ and $Q$ improves solution quality within neighborhoods but incurs greater per-step cost. While increasing $Q$ and $T$ improves generation quality, it also leads to longer run times. These results reveal a trade-off between computational efficiency and diagram accuracy.

**Visualization of Optimization Process.** To elucidate the iterative convergence behavior of the Monte Carlo algorithm, we present a visual analysis where in the intermediate states at each iteration are rendered as sequential snapshots, collectively forming a state transition diagram. This visualization offers an intuitive depiction of the diagram structure's progressive evolution toward the target configuration.

Complementing the structural visualization, we plot the loss values at each iteration in Figure 7 to characterize the algorithm's optimization dynamics. The resulting loss curve traces the objective function trajectory, revealing a consis-

*Table 3.* Efficienty results (Unit:%). Construct the training-free based on train set, average of the three fine-tuned Qwen2.5-7b on the test set, average of the three training-free based on test set. Three experiment of fine-tuned Qwen2.5-7b on the test set, three training-free experiments based on the test set.

| Method | $< 10s$ | $11s - 20s$ | $21s - 30s$ | $31s - 40s$ | $41s - 50s$ | $51s - 60s$ | $> 60s$ |
|---|---|---|---|---|---|---|---|
| Training-free (train set) | 75.51 | 5.53 | 1.98 | 5.61 | 3.04 | 1.95 | 6.30 |
| Training-free (test set) | 72.13 | 6.71 | 2.65 | 1.71 | 5.38 | 4.45 | 6.94 |
| Fine-tuned Qwen2.5-7b | 71.36 | 7.85 | 2.83 | 2.83 | 1.94 | 1.46 | 11.73 |

*Table 4.* Ablation study of filters and constraint. "w Filter" indicates the presence of filter configuration, while "w/o Filter" indicates the removal of filter configuration.

| Setting | LCI↓ | ADI↓ | Average | Time |
|---|---|---|---|---|
| *Filters* | | | | |
| w Filter | 0.962 | 1.747 | 1.337 | 13.379 |
| w/o Filter | 1.340 | 1.864 | 1.602 | 15.945 |
| *Constraint* | | | | |
| Full Constraint | 0.870 | 1.537 | 1.203 | 15.782 |
| Only Length | 0.651 | - | - | 2.844 |
| Only Angle | - | 1.577 | - | 7.477 |

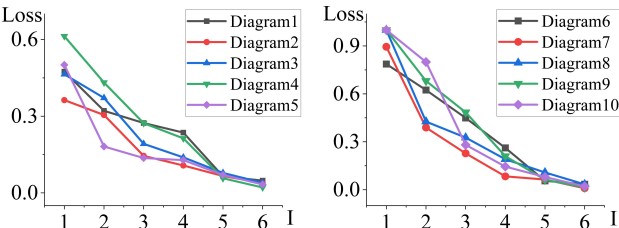

*Figure 7.* Loss curve during optimization.(The corresponding natural language and geometric diagrams can be found in Figure14). The geometry diagrams as shown in Figure15.

tent decline that aligns with increasing structural fidelity. This correlation indicates that the loss function effectively guides the search toward geometrically coherent solutions.

To verify the robustness of the autoformalization and visualize the convergence process during the coordinate solving phase, we conducted experiments **Randomness in autoformalization and coordinate solver** in the Appendix E.2.

## 6. Conclusion

In this work, we introduce GeoLoom, a novel framework designed to overcome the structural limitations of conventional text-to-image models in the geometric domain. By establishing GeoLingua, a generation-oriented formal language, and curating GeoNF, a high-quality paired dataset, we provide a principled foundation for aligning natural language with rigorous geometric construction. GeoLoom's two-stage architecture, decoupling logical autoformaliza-

tion from Monte Carlo-based spatial optimization, achieves a unique balance of mathematical precision and computational efficiency. Through extensive experiments, we show that GeoLoom significantly outperforms existing baselines, making it a promising solution for scalable, accurate diagram generation in educational contexts. This work paves the way for future research in symbolic-geometric modeling and the integration of structured formalism into multimodal generation tasks.

**Limitations.** While GeoLoom effectively automates 2D diagram generation, it has certain boundaries that offer avenues for future exploration. First, the current framework is strictly designed for planar Euclidean geometry. Extending the coordinate solver to handle 3D spatial constructions (e.g., prisms, spheres) or non-Euclidean surfaces would require a more complex manifold-based optimization logic. Second, the model's performance is tied to the clarity of the input text; highly ambiguous or nested linguistic descriptions can occasionally lead to errors in the autoformalization stage. Future work will focus on improving the system's robustness to diverse phrasing and expanding the GeoLingua vocabulary to support multi-dimensional geometry.

## Acknowledgments

This work was jointly supported by the National Natural Science Foundation of China (62437001) and the Fundamental Research Funds for the Central Universities (2253500001, 2243100004).

## Impact Statement

This work mainly utilizes large model autoformalization tasks and Monte Carlo algorithms to generate text to geometric diagrams. To protect privacy, all data and code will not be disclosed during the review stage. During the data collection process, manual annotators carefully review and remove all identifiable personal information, including name, school identification, and geographic location. In addition, potentially harmful or inappropriate content (such as offensive language) has been automatically filtered by the system. Therefore, the dataset used in this article only contains interaction content that is de identified and relevant

to teaching.

From a broader perspective, as multimodal models are increasingly deployed in classroom teaching, tutoring platforms, and remote learning environments, their ability to provide context sensitive and teaching compliant guidance has become one of the key factors determining the quality of education. Therefore, this framework not only facilitates model benchmark evaluation, but also promotes the deployment of LLMs in educational settings.

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

## A. Dataset

### A.1. Formal Language Framework

This chapter will provide a detailed introduction to GeoNF and the framework for defining formal languages (Section 3).

The formal language statements of the four module in GeoLingua are shown in the Table 5, Table 6, Table 7 and Table 8. Geometric names, length values, and angle values in formal language can be adjusted according to actual situations.

*Table 5.* Formal language definitions for "shapes" of GeoLingua.

| Natural Language | Formal Language |
|---|---|
| Circle /⊙, radius=r | `Circle(center_name,radius_value)` |
| Triangle | `Polygon(triangle_name)` |
| Parallelogram | `Parallelogram(parallelogram_name)` |
| Rhombus | `Rhombus(rhombus_name)` |
| Square | `Square(square_name)` |
| Rectangle | `Rectangle(rectangle_name)` |
| Trapezoid | `Polygon(trapezoid_name)` |
| Sector | `Sector(sector_name,the angle of sector,radius_value)` |
| String of circle | `StringOfCircle(circle_center,string_name)` |
| Circle inscribed polygon | `InscribedPolygon(circle_center,polygon_name)` |
| The inscribed circle of a polygon | `CircumscribedPolygon(circle_center,polygon_name)` |
| Other polygon | `Polygon(polygon_name)` |

### A.2. GeoNF Dataset Annotation

To ensure high-quality annotations, we implemented a multi-stage pipeline with three roles: (i) 10 mathematics-major annotators, (ii) a quality inspection team of 3 middle school teachers, and (iii) an Acceptance Team of 2 domain experts.

(1) Phase 1: Standardized Specification: The Acceptance Team defined a detailed annotation protocol covering problem statements, solution objectives, formal language (GeoLingua), and diagrams, and explained it to annotators and inspectors for consistent understanding.

(2) Phase 2: Data Collection and Curation: The annotator team collected geometry problems from middle school examinations over the past five years. An initial pool of approximately 5,000+ problems was obtained. The annotators then conducted manual filtering to remove duplicates and low-quality samples (e.g., incomplete statements or missing diagrams), resulting in a refined dataset of approximately 4,700 high-quality instances.

(3) Phase 3: Independent Annotation: We adopted a double-blind annotation scheme. The dataset was randomly divided into 5 subsets, each annotated independently by two annotators. The inspection team reviewed all annotations, and disputed cases were escalated to the Acceptance Team for arbitration.

(4) Phase 4: Iterative Verification and Final Validation: The inspection team conducted a pairwise review of the two annotations per instance. Consistent, correct samples were accepted; inconsistent or incorrect samples were re-annotated iteratively. Acceptance Team performed an independent audit on at least 10% of each batch of final dataset; batches below 98% accuracy were re-verified by the inspection team. This process ensured overall annotation accuracy more than 98%.

## B. Training-free Details

This section will provide a detailed introduction to prompt formats and validation-based filter of the training-free (Section 4.1).

### B.1. Prompt Formats

For the design of prompt, we have divided it into four parts: role, formal language framework, examples, and requirements. The content of each section is as follows:

*Table 6.* Formal language definitions for "depengence" of GeoLingua.

| Natural Language | Formal Language |
|---|---|
| Point at line segment | `PointAtLine(point_name,line_name,0)` |
| Proportional point of line segment | `PointAtLine(point_name,line_name,ratio_value)` |
| Intersection | `LineIntersect(line_name1,line_name2,intersection)` |
| Point on the arc | `PointAtArc(point_name,arc_name,arc_type,)` |
| Point P on the circle | `PointAtArc(point_name,circle_center,0)` |
| Extend the points on the line | `ExtensionLine(extension-line_name,point_name)` |
| Tangent of circle | `Tangent(circle_center,point of tangency,tangent)` |
| Make perpendicular line at point | `DoPerpendicular(point_name,line_name,foot)` |
| Point inside polygon | `PointInShape(point_namw,polygon_name,0)` |

*Table 7.* Formal language definitions for "length constraint" of GeoLingua.

| Natural Language | Formal Language |
|---|---|
| Segment length | `Length(line_name,length_value)` |
| Line segment ratio | `LengthRatio(line_name1,line_name2,Ratio(value1,value2))` |
| Arc ratio | `ArcRatio(arc_name1,arc_name2,Ratio(value1,value2))` |
| Line segment relationship | `LengthAddandSub((left side ),relation,(right side))` |
| Perimeter of polygon | `LengthAddandSub((lines_name of polygon),equal,perimeter)` |
| Distance from point to line | `PointLineDistance(point_name,line_name,distance_value)` |
| Connect line segment | `ConnectPoints(line_name)` |

- **Role:** You are a professional formal language translation assistant capable of accurately converting natural language descriptions (natural language) of geometry problems into structured formal language (formal language). The conversion rules from natural language to formal language are as follows:

- **Formal Language Framework:** A formal language framework in the form of markdown.

- **Examples:** Listed the diverse geometric natural language and formal language.

- **Requirements:** You must strictly follow the conversion rules defined above and special conversion rules for the conversion. The final output JSON format is as follows : "JSON data format". The natural language description entered by the user is as follows:" Among them, the JSON data format is: "shapes": [], "dependence": [], "length constraint": [], "angle constraint": []

## B.2. Validation-based Filter

There is an illusion problem with large models. In order to prevent the model from outputting undefined formal statements, we set validation-based filter to ensure that the formal language content meets the defined requirements. Validation-based filters are mainly designed to prevent tampering or the addition of formal language statements to the Large Language model. We have set up filters three times. If it does not meet the requirements of formal language, the prompt will be reloaded until all three chances are exhausted. If required, the formal language will be outputted.

## C. Method of Coordinate Solver

In this section, we will briefly introduce the algorithm of coording solver.

### C.1. Monte Carlo-based Algorithm

In this section, We have provided a detailed introduction to the implementation algorithm of a coordinate solver driven by Monte Carlo (Monte Carlo-based Algorithm of the Section 4.2). The specific algorithm process is shown in Algorithm 1.

*Table 8.* Formal language definitions for "angle constraint" of GeoLingua.

| Natural Language | Formal Language |
|---|---|
| ‖ | `Parallel(line_name1,line_name2,0)` |
| ⊥ | `Perpendicular(line_name1,line_name2,90)` |
| Degree of angle | `Angle(angle_name,degree_value)` |
| Trigonometric function | `TriFunction(function,angle_name,value)` |
| Angle ratio | `AngleRatio(angle_name1,angle_name2,Ratio(value1,value2))` |
| Angle relationship | `AngleAddandSub((left side),relation,(right side))` |

## C.2. Geometric Diagram Render Algorithm

This section will provide a detailed introduction to geometric rendering algorithms(Geometric Diagram render of the Section 4.2).

We use the regular matching method to extract the points identifiers $P$ and all line segment representations $L$ in the geometric diagram from the formal language. By solving the coordinates, we obtained the coordinates $P = \{p_1(x_1, y_1), \ldots, p_m(x_m, y_m)\}$ of each point. In order to better present the geometric diagram, we designed an adaptive canvas. The adaptive canvas algorithm is as Algorithm 2

# D. Experiment result

In this section, we will briefly introduce the formal language corresponding to baseline methods, geometric diagrams in qualitative evaluation, limitation of CLIP in geometric giagrams, and efficiency results in quantitative evaluation (Section 5.1).

## D.1. Baseline Methods:

In the comparative experiment, we have compared our method with the following baselines:

1) **AutomaTiKZ:** AutomaTikZ is an abstract graphics language based on vector drawing, which describes geometric shapes, charts, scientific diagrams, etc. through code, compiles and outputs scalable vector graphics. We choose the TiKZ-LLaMa7b model trained on LLaMa as base model.

2) **SeeDream:** Basic model for bilingual image generation in Chinese and English

3) **Penrose:** Penrose describes geometric and mathematical relationships using the Substance language, and then automatically generates a diagram that satisfies these constraints through optimization. However, its drawback is that users must first learn and manually specify this relatively complex description language.

4) **GeoGebra:** GeoGebra relies on manual diagram construction, which can ensure high visual quality but requires substantial human effort and time.

The AutomaTiKZ and SeeDream methods perform fully automatic geometric diagram generation, Penrose and GeoGebra require human intervention. On standard benchmark datasets(GeoNF), we compare our approach with AutomaTiKZ and SeeDream and show that it can generate correct diagrams both rapidly and reliably. On more challenging datasets such as IMO-style problems, we compare our method with Penrose and GeoGebra. Although these baselines are able to produce high-quality diagrams, under comparable visual quality our method is faster and does not require any manual involvement.

## D.2. Limitations of CLIP

Our evaluation already includes two specialized, geometry-based metrics: Length Consistency Index (LCI) and Angle Deviation Index (ADI), which quantify the deviation between generated diagram and target diagram based on constraints (e.g., lengths and angles). LCI and ADI are coordinate-based metrics designed to measure precise mathematical alignment. Baselines (Seedream3.0, AutomaTikZ) generate either raw pixels or non-structured TikZ code without explicit coordinate constraints, lacking the structured metadata required to calculate these geometric indices. This lack of "structural awareness"

---

**Algorithm 1** Monte Carlo-based algorithm

---

**Input:** Textual description $T$
**Output:** Geometric diagram $S_{global}$
Initialize geometric diagram $S^0 = (P_f^0, P_d^0, \mathcal{C})$ and Calculate the initial loss value $\mathcal{L}(S^0)$;
  Define the global optimal solution as the initial state $S_{global} = S^0$;

**while** $t < T$ *or* $\mathcal{L}(S)_{\max} > \alpha$ **do**
    for the (t-1)-th iteration result: $S^{t-1} = (P_f^{t-1}, P_d^{t-1}, \mathcal{C})$ ;
    **while** $q < Q$ **do**
      $S^q = S^{t-1}$;
      $\mathcal{L}(S)_{best} = \mathcal{L}(S^q)_{max}$;
      Monte Carlo perturbation free point $P_f^q$;
      $\mathcal{L}(S^q)_{max} = \text{evaluate}(S^q, P_f^q, P_d^q)$;

      **if** $\mathcal{L}(S^q)_{max} < \mathcal{L}(S)_{best}$ **then**
        Update local optimal solution: $\mathcal{L}(S)_{best} = \mathcal{L}(S^q)_{max}$;
        $S_{local} = S^q$ ;

      **end**
    **end**
    Update the dependency points based on the local optimal solution: $S^{(t)} = S_{local}$; $P_d^{(t)} = \Phi(P_f^t, \mathcal{C})$;
    $\mathcal{L}(S^t)_{max} = \text{evaluate}(S^t, P_f^t, P_d^t)$;
    **if** $\mathcal{L}(S^t)_{max} < \mathcal{L}(S)_{best}$ **then**
      Update the global optimal solution: $\mathcal{L}(S)_{best} = \mathcal{L}(S^t)_{max}$;
      $S_{global} = S^t$ ;

    **end**
    **return** $S_{global}$;
**end**

---

in baselines is precisely the limitation our GeoLoom framework aims to solve. To provide a side-by-side comparison, we conducted experiments on Figure4 using the CLIP Score to measure **image-text alignment** and **image similarity**. As can be observed in Figure4, the outputs generated by baselines exhibit significant deviations from the ground truth but still achieve high CLIP scores. In contrast, our generated geometric results, which are visually more consistent with the ground truth, receive lower CLIP scores. The findings indicate that CLIP is unsuitable for evaluating geometric diagram quality.

*Table 9.* Proving the Limitations of CLIP in Geometric Diagrams Based on Figure4. (image-text alignment | image similarity)

| Row | Ours (training-free) | | Ours (fine-tuned) | | SeeDream3.0 | | AutomaTiKZ | |
|---|---|---|---|---|---|---|---|---|
| The first row | 0.2548 | 0.8135 | 0.2548 | 0.8135 | 0.2736 | 0.9207 | 0.2824 | 0.8652 |
| The second row | 0.2463 | 0.7390 | 0.2463 | 0.7390 | 0.2729 | 0.8784 | 0.2766 | 0.9267 |
| The third row | 0.2828 | 0.9249 | 0.2870 | 0.8953 | 0.3130 | 0.9184 | 0.3114 | 0.8798 |
| The fourth row | 0.2251 | 0.8960 | 0.2230 | 0.9241 | 0.2129 | 0.8862 | 0.2282 | 0.9209 |

### D.3. Qualitative Evaluation Experiment

The additional results in the qualitative evaluation experiment are shown in Figure 8. (The other resultds of qualitative evaluation is shown in the Figure 4 of Section 5.1). The formal languages corresponding to the geometric diagrams are shown as Figure 12. The natural-language descriptions corresponding to Figure 5 in Section 5.1 are shown in the Figure 11.

**Algorithm 2** Geometric Diagram Render Algorithm

**Input:** Geometric diagram $S_{global}$.
**Output:** Image of geometric diagram.
Extract all points coordinate $P = p_1(x_1, y_1), \ldots, p_m(x_m, y_m)$ from $S_{global}$.

Regularly parsing points $P$ and lines $L$ from formal languages.

**step1: Calculate coordinate offsets**.
  $x_f$ = -min(x) if $\min(x) < 0$ else 0;
  $y_f$ = -min(y) if $\min(y) < 0$ else 0;

**step2: Apply coordinate translation**.

**if** $x_f \neq 0$ *or* $y_f \neq 0$ **then**
    **for** *each point* $p \in P$ **do**
      |  $x \leftarrow x + x_f; y \leftarrow y + y_f$;
    **end**
    Update coordinate sets: $P = p_1(x_1, y_1), \ldots, p_m(x_m, y_m)$;
**end**
**step3: Determine canvas size** $C_{size}$ .

$M_{coord} \leftarrow \max(\max(x), \max(y))$;
**if** $M_{coord} > C_{size}$ **then**
  |  $C_{size} \leftarrow (\lfloor M_{coord}/50 \rfloor + 1) \times 50$ (50 is used to ensure that the canvas is an integer multiple of 50);
**end**
Draw a geometric diagram with lines $L$ and adjusted coordinates $P$ on a new canvas $C_{size}$.
  **return** Geometric diagram;

| Ground Truth | | Ours | | Without Prior Knowledge | | | With Prior Knowledge | | |
|---|---|---|---|---|---|---|---|---|---|
| Natural Language | Origin-img | Training-free | Fine-tuned | AutomaTiKZ | Seedream3.0 | Seedream5.0 | AutomaTiKZ | Seedream3.0 | Seedream5.0 |
| In diamond shapes ABCD, AC and BD intersect at points O, E, and F, which are the midpoints of OA and OC, respec-tively, Connect DF and BE. | | | | | | | | | |
| In square ABCD, the side length of square ABCD is 4, connecting AC and BD at point O, extending DC to point E, connecting AE, ∠CAE=15°. | | | | | | | | | |

*Figure 8.* Additional result of quantitative evaluation. The baseline methon are AutomaTikz (base LLaMa-7b) and SeeDream model.

## D.4. Efficiency Results

In the efficiency experiment (Efficiency result of the Section 5.1), three times experiments were conducted on the fine-tuned Qwen2.5-7b model and the training-free DeepSeek-V3, and the time consumption of the three times experiments and the The average time are shown in Table 3 . The three experiments in Table 3 indicate that the majority of geometric graph generation takes less than 10 seconds. The results of the time proportions of the two methods indicate that the efficiency of the coordinate parser is highly stable.

## E. Experiment Analysis

In this section, We will supplement necessary experimental analysis for Section 5.2).

### E.1. Failure Case of GeoLoom

- Topological of formal language: For a very small number of natural-language descriptions that are not written in a standard form, the automatic formalization process can produce errors in the topological ordering. As shown by the

example on the right in Figure 9, the natural-language description uses DE ∥ AB, DF ∥ AC, while the standardized description should be DE ∥ BA, DF ∥ CA. Since the topological order is not corrected during formalization, the geometric diagram cannot be constructed correctly, and the evaluation metrics become extremely abnormal. Such cases account for 3% of the test set.

- Local minima in Monte Carlo coordinate optimization: Since the geometric constraint objective is inherently highly non-convex (especially when multiple angle, ratio, collinearity, and concyclicity constraints are involved), the optimization process is inevitably prone to local minima. The local minimum solution is characterized by evaluation metrics that are close but do not meet the threshold, and geometric diagrams that visually resemble the target image but do not meet the required details. In the middle example of Figure 9, the segment ratio constraint is correctly specified in the formal language. However, during random optimization, the target ratio of 2:1 cannot be reached, causing the procedure to converge to a local minimum.

- Overlapping problem: When two points are too close to each other, a quasi-overlapping issue arises. Although such cases still satisfy the quantitative evaluation metrics, they are visually unsatisfactory for practical use, as shown in Figure 9. These failure cases account for 3% of the test dataset.

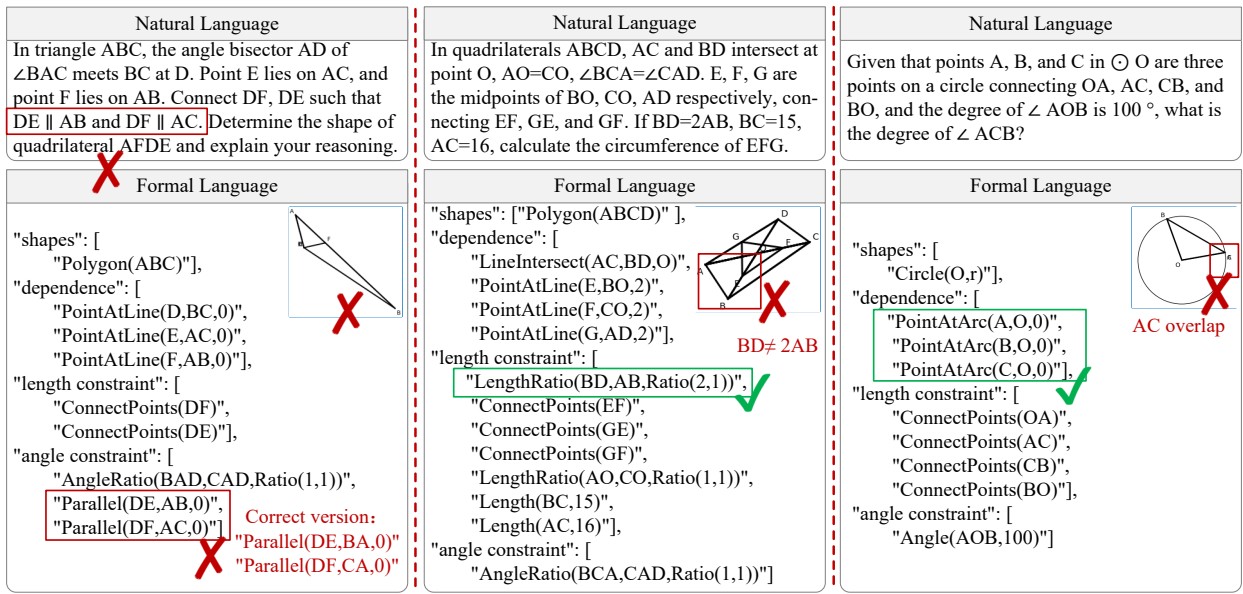

*Figure 9.* Analysis of Failure Cases. On the left is a case of topological of formal language. In the middle is a case of local minima in Monte Carlo coordinate optimization, and on the right is a case of overlapping problem.

### E.2. Randomness in Autoformalization and Coordinate Solver

Translating natural language into formal geometric representations inherently involves ambiguity, resulting in multiple formally distinct yet semantically equivalent expressions. This variation arises from the probabilistic nature of language models, which produce diverse outputs given identical inputs. Despite these differences, all generated formalizations preserve the underlying geometric semantics and constraints. Consequently, when processed by the downstream diagram generation system, they yield mathematically valid and structurally correct diagrams. This demonstrates the robustness of the autoformalization process in maintaining semantic fidelity amid linguistic variability.

Beyond autoformalization, stochasticity also occurs in the coordinate solving stage. Our system's Monte Carlo–based solver introduces randomness in point coordinate generation. Although each sampling iteration may produce different coordinates, these variations remain bounded within predefined tolerance thresholds that guarantee geometric consistency. Figure 10 illustrates these two controlled sources of variation, highlighting the difference between autoformalization, list the geometric diagram generated by random sampling within the tolerance threshold.

| Natural Language & Difference of formal language | Training-free | Fine-tuned | | | | |
|---|---|---|---|---|---|---|
| | | LLaMa3.1-8b | LLaMa3.2-3b | LLaMa3-8b | Qwen2.5-7b | Qwen2.5-14b |
| In rhombus ABCD, connect AC and BD, with diagonal lines AC and BD intersecting at point O, AC=4, BD=8, point E on the edge Of AD, **3AE=AD**. Connect BE, BE intersects AC at point M. 
 **(I) 3AE=AD :** 
 *LengthRatio(AE,AD,Ratio(1,3))* 
 **(II)2AE=ED:** 
 *LengthRatio(AE,ED,Ratio(1,2))* |  
 (I) |  
 (I) |  
 (I) |  
 (I) |  
 (II) |  
 (I) |
| In rectangles ABCD, point E is on line segment CD, point F is on the extension of line segment AB, connecting EF to line segment BC at point G, and connecting BD, **DE=BF=2**. 
 **(I)DE=BF=2:** 
 *Length(DE,2), Length(BF,2)* 
 **(II)DE=2,DE:BF=1:1:** 
 *Length(DE,2),* 
 *LengthRatio(DE,BF,Ratio(1,1))* |  
 (I) |  
 (II) |  
 (I) |  
 (II) |  
 (II) |  
 (I) |
| In triangle ABC, AB=AC，**make AD perpendicular to BC at point D**，If ∠BAC=70 °. 
 **(I)make AD⊥BC:** 
 *DoPerpendicular(A,BC,D)* 
 **(II)D at line BC, AD⊥BC:** 
 *PointAtLine(D,BC,0),* 
 *Perpendicular(AD,BC,90)"* |  
 (I) |  
 (II) |  
 (I) |  
 (I) |  
 (II) |  
 (I) |

*Figure 10.* Randomness in autoformalization and coordinate. Highlight the specific formal language that is different in fine-tuned and training-free methods.

In the randomness in autoformalization and coordinate solver experimental (Randomness in autoformalization and coordinate solver of Section 5.2), The experimental results are as Figure 10. The formal language corresponding to the geometric diagram is as Figure13.

| Natural lanuage | Formal language | | | |
|---|---|---|---|---|
| | shapes | dependence | length constraint | angle constraint |
| Let ABCD be a cyclic quadrilateral with circumcircle O, angle ABC = 120°, and angle CDA = 60°. Extend CD beyond D to a point J such that DJ = AD. Extend BC beyond C to a point K such that CK = BC. Join AJ and KJ. Extend CB beyond B to a point E and join AE, such that angle AEB = 90°. Let F, G, H, I be the midpoints of AB, BC, CD, DA, respectively. Join FG, GH, HI, IF, FH, and GI. Let FH and GI intersect at P, and let BA and DC intersect at Q. Prove that FGHI is a parallelogram. | "Circle(O,r)", "InscribedPolygon(O,ABCD)" | "ExtensionLine(BC,K)", "ExtensionLine(CD,J)", "ExtensionLine(CB,E)", "PointAtLine(F,AB,2)", "PointAtLine(G,BC,2)", "PointAtLine(H,CD,2)", "PointAtLine(I,DA,2)", "LineIntersect(FH,GI,P)", "LineIntersect(AB,DC,Q)" | "LengthRatio(DJ,AD,Ratio(1,1))", "LengthRatio(CK,BC,Ratio(1,1))", "ConnectPoints(AE)", "ConnectPoints(FG)", "ConnectPoints(GH)", "ConnectPoints(HI)", "ConnectPoints(IF)", "ConnectPoints(AJ)", "ConnectPoints(KJ)" | "Angle(ABC,120)", "Angle(ADC,60)", "Angle(AEB,90)" |
| Let ABC be a triangle inscribed in circle O, with AB = BC. Point F lies on line BC such that CF = BC. Point E lies on ray BA beyond A such that BA = AE. Point G is the midpoint of AC. Join EF, EG, and GF. Extend CB to a point H, and extend AB to a point I. Join HE, HG, HI, and FI. The construction is such that HI is parallel to EF. Prove that quadrilateral HIEF is an isosceles trapezoid. | "Circle(O,r)", "InscribedPolygon(O,ABC)" | "ExtensionLine(BC,F)", "ExtensionLine(BA,E)", "ExtensionLine(CB,H)", "ExtensionLine(AB,I)", "PointAtLine(G,AC,2)" | "LengthRatio(AB,BC,Ratio(1,1))", "LengthRatio(CF,BC,Ratio(1,1))", "LengthRatio(AE,BA,Ratio(1,1))", "LengthRatio(BH,BI,Ratio(1,1))", "ConnectPoints(EG)", "ConnectPoints(FG)", "ConnectPoints(HG)", "ConnectPoints(IG)", "ConnectPoints(EF)", "ConnectPoints(HE)", "ConnectPoints(HI)", "ConnectPoints(IF)" | "Parallel(HI,EF,0)" |
| In a rhombus ABCD, the diagonals AC and BD intersect at O. Let E be the midpoint of CD, and G the midpoint of AB. Extend segment OE beyond E to a point F such that OE = EF. Extend segment OG beyond G to a point H such that OG = GH. Prove that HDBF is a parallelogram, and that quadrilaterals AHBO and DOCF are squares. | "Rhombus(ABCD)" | "LineIntersect(AC,BD,O)", "PointAtLine(E,CD,2)", "PointAtLine(G,AB,2)", "ExtensionLine(OE,F)", "ExtensionLine(OG,H)" | "LengthRatio(AB,CD,Ratio(1,1))", "LengthRatio(BC,AD,Ratio(1,1))", "LengthRatio(AB,BC,Ratio(1,1))", "LengthRatio(OE,EF,Ratio(1,1))", "LengthRatio(OG,GH,Ratio(1,1))", "ConnectPoints(DF)", "ConnectPoints(CF)", "ConnectPoints(AH)", "ConnectPoints(BH)", "ConnectPoints(DH)", "ConnectPoints(BF)" | "Parallel(AB,DC,0)", "Parallel(AD,BC,0)", "Angle(ABC,100)" |
| In circle O, triangle ABC is inscribed and satisfies AB = AC. Through B, draw BE perpendicular to AC with foot E on AC. Through A, draw AD perpendicular to BC with foot D on BC. Through E, draw EK perpendicular to BC with foot K on BC. Join O and E; suppose OE is perpendicular to AD. Lines BE and AD intersect at H, and it is given that AH = 2·OD. Point G lies on segment AD. Join BG, CG, and GE. | "Circle(O,r)", "InscribedPolygon(O,ABC)" | "PointAtLine(E,AC,0)", "PointAtLine(D,BC,0)", "PointAtLine(K,DC,0)", "PointAtLine(G,AD,0)", "LineIntersect(BE,AD,H)" | "LengthRatio(AB,AC,Ratio(1,1))", "LengthRatio(AH,OD,Ratio(2,1))", "ConnectPoints(BG)", "ConnectPoints(EG)", "ConnectPoints(EO)", "ConnectPoints(GC)" | "Perpendicular(AD,BC,90)", "Perpendicular(BE,AC,90)", "Perpendicular(EO,AD,90)", "Perpendicular(EK,BC,90)" |
| An equilateral triangle ABC is inscribed in circle O, and angle ABC = 90°. Chord DE lies on circle O, and point P is a point on segment BC. Extend BA beyond A to a point R, and join PR. Let PR intersect AC at Q. Join BE, RE, DR，DP and DQ, with DQ perpendicular to AC, DR perpendicular to AB and DP be perpendicular to BC. | "Circle(O,r)", "InscribedPolygon(O,ABC)", "StringOfCircle(O,DE)" | "PointAtLine(P,BC,0)", "ExtensionLine(BA,R)", "LineIntersect(RP,AC,Q)" | "PointLineDistance(O,DE,0)", "ConnectPoints(DR)", "ConnectPoints(RE)", "ConnectPoints(RP)", "ConnectPoints(DQ)", "ConnectPoints(DP)", | "Angle(ABC,90)", "Perpendicular(DR,AB,90)", "Perpendicular(DQ,AC,90)", "Perpendicular(DP,BC,90)" |
| In circle O, triangle ABC is inscribed with angle BAC = 80°. KE is a diameter of circle O. Another diameter AD bisects angle BAC. Point F is the midpoint of BC, and point G is the midpoint of segment DK. Join DK, FG, GO, and FO. It is given that FG = FO. | "Circle(O,r)", "InscribedPolygon(O,ABC)", StringOfCircle(O,KE)", "StringOfCircle(O,AD)" | "PointAtLine(F,BC,0)", "PointAtLine(G,DK,0)" | "PointLineDistance(O,KE,0)", "PointLineDistance(O,AD,0)", "LengthRatio(FG,FO,Ratio(1,1))", "ConnectPoints(FG)", "ConnectPoints(GO)", "ConnectPoints(FO)", "ConnectPoints(DK)" | "Angle(BAC,80)", "AngleRatio(BAD,CAD,Ratio(1,1))" |

*Figure 11.* The formal language corresponding to natural language in Figure5.

| Natural lanuage | Formal language | | | |
|---|---|---|---|---|
| | shapes | dependence | length constraint | angle constraint |
| The triangle ABC is an inscribed equilateral triangle of circle O, point O is the center, points D and E are on sides AC and AB respe c tively, con - necting OD and OE, DA=BE. | "Circle(O,r)", "InscribedPolygon(O,ABC)" | "PointAtLine(D,AC,0)", "PointAtLine(E,AB,0)"], | "ConnectPoints(OD)", "ConnectPoints(OE)", "LengthRatio(AB,BC,Ratio(1,1))", "LengthRatio(AB,AC,Ratio(1,1))", "LengthRatio(DA,EB,Ratio(1,1))" | [] |
| In triangle ABC, points D, E, F are respectively the midpoints of AB, AC, and BC, Connecting DE, EF, and DF. Points P, M, and N are respectively the midpoints of DE, DF, and EF, connecting PM, PN, and MN. | "Polygon(ABC)" | "PointAtLine(D,AB,2)", "PointAtLine(E,AC,2)", "PointAtLine(F,BC,2)", "PointAtLine(P,DE,2)", "PointAtLine(M,DF,2)", "PointAtLine(N,EF,2)"], | "ConnectPoints(DE)", "ConnectPoints(EF)", "ConnectPoints(DF)", "ConnectPoints(PM)", "ConnectPoints(PN)", "ConnectPoints(MN)" | [] |
| In the four sides ABCD, ∠BAD=120◦, ∠ABC=∠ADC =90◦, find points M and N on BC and CD respectively, and connect AM and AN. | "Polygon(ABCD)" | "PointAtLine(M,BC,0)", "PointAtLine(N,CD,0)" | "ConnectPoints(AM)", "ConnectPoints(AN)", | "Angle(BAD,120)", "Perpendicular(BA,BC,90)", "Perpendicu-lar(DA,DC,90)" |
| I n rectangle ABCD, point E is on edge AB, connected to DE, D E is the bisector of ∠ADC. Point F is on the extension of DE, connected to BF, ∠BFE = 90◦, connected to AF, CF, CF, and AB intersecting with G. | "Rectangle(ABCD) | "PointAtLine(E,AB,0)", "ExtensionLine(DE,F)", "LineIntersect(CF,AB,G)" | "LengthRatio(AD,BC,Ratio(1,1))", "LengthRatio(AB,DC,Ratio(1,1))", "ConnectPoints(DE)", "ConnectPoints(BF)", "ConnectPoints(AF)", "ConnectPoints(CF)" | "Perpendicular(AB,AD,90)", "Perpendicular(DA,DC,90)", "Perpendicular(FB,FE,90)", "AngleRatio(ADE,CDE, Ratio(1,1))" |
| I n diamond shapes ABCD, AC and BD intersect at points O, E, and F, which are the midpoints of OA and OC, respectively, c onnect DF and BE | "Rhombus(ABCD)" | "LineIntersect(AC,BD,O)", "PointAtLine(E,OA,2)", "PointAtLine(F,OC,2)" | "LengthRatio(AB,CD,Ratio(1,1))", "LengthRatio(BC,AD,Ratio(1,1))", "LengthRatio(AB,BC,Ratio(1,1))", "ConnectPoints(AC)", "ConnectPoints(BD)", "ConnectPoints(DF)", "ConnectPoints(BE)" | "Parallel(AB,DC,0)", "Parallel(AD,BC,0)" |
| In square ABCD, the side length of square ABCD is 4, connecting AC and BD at point O, extending DC to point E, connecting AE, ∠CAE=15◦ | "Square(ABCD)" | "LineIntersect(AC,BD,O)", "ExtensionLine(DC,E)" | "LengthRatio(AB,AD,Ratio(1,1))," LengthRatio(AB,BC,Ratio(1,1))", "Length(AB,4)", "ConnectPoints(AE)" | "Perpendicular(AB,AD,90)", "Perpendicular(DA,DC,90)", "Angle(CAE,15)" |

*Figure 12.* The formal language corresponding to natural language in Figure4 and Figure8.

| Natural lanuage | Formal language | | | |
|---|---|---|---|---|
| | shapes | dependence | length constraint | angle constraint |
| In rhombus ABCD, connect AC and BD , with diagonal li- nes AC and BD intersecting at point O, AC=4, BD=8, point E on the edge Of AD, 3AE=AD. BE intersects AC at point M. | "Rhombus(ABCD)" | "LineIntersect(AC,BD,O)", "PointAtLine(E,AD,0)", "LineIntersect(BE,AC,M)" | "ConnectPoints(AC)", "ConnectPoints(BD)", "Length(AC,4)", "Length(BD,8)", "LengthRatio(AE,AD,Ratio(1,3))", "LengthRatio(AB,CD,Ratio(1,1))", "LengthRatio(BC,AD,Ratio(1,1))", "LengthRatio(AB,BC,Ratio(1,1))" | "Parallel(AB,DC,0)", "Parallel(AD,BC,0)" |
| In rectangles ABCD, point E is on line segment CD, point F is on the extension of line seg - ment AB, connecting EF to line segment BC at point G, connecting BD, DE=BF=2 | "Rectangle(ABCD)" | "PointAtLine(E,CD,0)", "ExtensionLine(AB,F)", "LineIntersect(EF,BC,G)" | "ConnectPoints(EF)", "ConnectPoints(BD)", "Length(DE,2)", "Length(BF,2)" | [] |
| In triangle ABC, AB=AC, make AD perpendicular to BC at point D，∠BAC=70 °. | "Polygon(ABC)" | "DoPerpendicular(A,BC,D)" | "LengthRatio(AB,AC,Ratio(1,1))" | "Angle(BAC,70)" |

*Figure 13.* The formal language corresponding to natural language of Figure10.

| Natural lanuage | Formal language | | | |
|---|---|---|---|---|
| | shapes | dependence | length constraint | angle constraint |
| In △ ABC, ∠ ACB=90 °, ∠ ABC=30 °, AD bisect ∠BAC , A D passing through A intersects BC at D, DE ⊥ AB , perpendicular foot is E, DE=1. | "Polygon(ABC)" | "PointAtLine(D,AC,0)", "PointAtLine(E,AB,0)"], | "Length(AB,8)", "Length(BC,12)", "ConnectPoints(BD)", "ConnectPoints(AE)", "ConnectPoints(CE)" | "Perpendicular(BA,BC,90)", "AngleRatio(ABD,BCE, Ratio(1,1)" |
| In △ ABC, ∠ ABC=90 °, AB=8, BC=12, D is a moving point on the AC side, connecting BD. E is a moving point on BD, connecting AE and CE, ∠ ABD=∠ BCE. | "Polygon(ABC)" | "PointAtLine(D,BC,0)", "DoPerpendicular(D,AB,E)" | "Length(DE,1.5)", "Length(BD,3)" | "Perpendicular(CA,CB,90)", "AngleRatio(BAD,DAC, Ratio(1,1)" |
| In △ ABC, ∠ ACB=90 °, AD divides ∠ BAC equally and intersects BC with D, DE is perpendicular to AB with E. DE=1.5, BD=3. | "Polygon(ABC)" | "PointAtLine(D,BC,0)", "DoPerpendicular(D,AB,E)" | "Length(DE,1.5)", "Length(BD,3)" | "Perpendicular(CA,CB,90)", "AngleRatio(BAD,DAC, Ratio(1,1)" |
| In the parallelogram ABCD, BD is the diagonal, BD=CD, ∠ BCD=70 °, AE is perpendicular to BD at point E. | "Parallelogram(ABCD)" | "DoPerpendicular(A,BD,E)" | "LengthRatio(AB,CD,Ratio(1,1))", "LengthRatio(BC,AD,Ratio(1,1))", "LengthRatio(BD,CD,Ratio(1,1))" | "Parallel(AB,DC,0)", "Parallel(AD,BC,0)", "Angle(BCD,70)" |
| In parallelogram ABCD, the diagonal lines AC and BD intersect at point O, AB ⊥ AC. Point H is on the BD, AH ⊥BD, AB=2, BC=$2sqrt{3}$. | "Parallelogram(ABCD)" | "LineIntersect(AC,BD,O)", "PointAtLine(H,BD,0)", | "LengthRatio(AB,CD,Ratio(1,1))", "LengthRatio(BC,AD,Ratio(1,1))", "LengthRatio(BD,CD,Ratio(1,1))", "Length(AB,2)" , "Length(BC,$2sqrt{3}$)" , | "Parallel(AB,DC,0)", "Parallel(AD,BC,0)", "Perpendicular(AH,BD,90)", |
| In triangle ABC, ∠ BCA=90 °, AC=3, ∠ ABC=30 °, point P is the moving point on the side of BC, connecting AP. | "Polygon(ABC)" | "PointAtLine(P,BC,0)" | "Length(AC,3)", "ConnectPoints(AP)" | "Perpendicular(CB,CA,90)", "Angle(ABC,30)" |
| AB is the diameter of ⊙ O, and the string CD ⊥ AB connects OC and BD at point E. ∠ AOC=110 °. | "Circle(O,r)", "StringOfCircle(O,AB)", "StringOfCircle(O,CD)" | "LineIntersect(AB,CD,E)" | "PointLineDistance(O,AB,0)", "ConnectPoints(OC)", "ConnectPoints(BD)" | "Perpendicular(AB,CD,90)", "Angle(AOC,110)" |
| In rhombus ABCD, the diagonals AC and BD intersect at point O, where OA=1 and OB=2. | "Rhombus(ABCD)" | "LineIntersect(AC,BD,O)" | "LengthRatio(AB,CD,Ratio(1,1))", "LengthRatio(BC,AD,Ratio(1,1))", "LengthRatio(AB,BC,Ratio(1,1))", "Length(OA,1)", "Length(OB,2)" | "Parallel(AB,DC,0)", "Parallel(AD,BC,0)" |
| In the rhombus ABCD, E, F, G, and H are the midpoints of AB, BC, CD, and DA, respectively. AB=6, ∠ ABC=60 °, connecting EF, FG, GH, and HE. | "Rhombus(ABCD)" | "PointAtLine(E,AB,2)", "PointAtLine(F,BC,2)", "PointAtLine(G,CD,2)", "PointAtLine(H,DA,2)" | "LengthRatio(AB,CD,Ratio(1,1))", "LengthRatio(BC,AD,Ratio(1,1))", "LengthRatio(AB,BC,Ratio(1,1))", "Length(AB,6)", "ConnectPoints(EF)","ConnectPoints(FG)","ConnectPoints(GH)", "ConnectPoints(HE)" | "Parallel(AB,DC,0)", "Parallel(AD,BC,0)", "Angle(ABC,60)" |
| In the rhombus ABCD, passing through point B is marked as BE ⊥ AD, BF ⊥ CD, with vertical feet at points E and F, connecting BD and extending to G, DG=BD, connecting EG and FG, AE=DE. | "Rhombus(ABCD)" | "DoPerpendicular(B,AD,E)", "DoPerpendicular(B,CD,F)", "ExtensionLine(BD,G)" | "LengthRatio(AB,CD,Ratio(1,1))", "LengthRatio(BC,AD,Ratio(1,1))", "LengthRatio(AB,BC,Ratio(1,1))", "LengthRatio(DG,BD,Ratio(1,1)), "LengthRatio(AE,DE,Ratio(1,1))", "ConnectPoints(EG)", "ConnectPoints(FG)" | "Parallel(AB,DC,0)", "Parallel(AD,BC,0)" |

*Figure 14.* The formal language corresponding to natural language of Figure7.

| Natural Language | Initial state | Optimization process | | | | Final state |
|---|---|---|---|---|---|---|
| In △ ABC, ∠ ACB=90 °, ∠ ABC=30 °, the angle bisector AD of ∠ BAC passing through point A intersects BC at point D, and the perpendicular foot of DE ⊥ AB passing through point D is E, DE=1. | | | | | | |
| In △ ABC, ∠ ABC=90 °, AB=8, BC=12, D is a moving point on the AC side, connecting BD. E is a moving point on BD, connecting AE and CE, ∠ ABD=∠ BCE. | | | | | | |
| In △ ABC, ∠ ACB=90 °, AD divides ∠ BAC equally and intersects BC with D, DE is perpendicular to AB with E. DE=1.5, BD=3 | | | | | | |
| In the parallelogram ABCD, BD is the diagonal, BD=CD, ∠ BCD=70 °, AE is perpendicular to BD at point E. | | | | | | |
| In parallelogram ABCD, the diagonal lines AC and BD intersect at point O, AB ⊥ AC. Point H is on the BD, AH⊥BD ，AB=2, BC=$2sqrt {3}$. | | | | | | |
| In triangle ABC, ∠ BCA=90 °, AC=3, ∠ ABC=30 °, point P is the moving point on the side of BC, connecting AP. | | | | | | |
| AB is the diameter of ⊙ O, and the string CD ⊥ AB connects OC and BD at point E. ∠ AOC=110 °. | | | | | | |
| In rhombus ABCD, the diagonals AC and BD intersect at point O, where OA=1 and OB=2. | | | | | | |
| In the rhombus ABCD, E, F, G, and H are the midpoints of AB, BC, CD, and DA, respectively. AB=6, ∠ ABC=60 °, connecting EF, FG, GH, and HE. | | | | | | |
| In the rhombus ABCD, passing through point B is marked as BE ⊥ AD, BF ⊥ CD, with vertical feet at points E and F, connecting BD and extending to G, DG=BD, connecting EG and FG, AE=DE. | | | | | | |

*Figure 15.* Visualization of optimization process of geometric diagrams.

