# OpenReview forum: "GeoLoom: High-quality Geometric Diagram Generation from Textual Input"
_ICML.cc/2026/Conference — ICML 2026 regular_

### Official Review · Reviewer_iYNY · 2026-03-06

**Soundness:** 2
**Presentation:** 2
**Significance:** 3
**Originality:** 3
**Overall Recommendation:** 4
**Confidence:** 4

**Summary:**

This paper aims to generate geometric diagrams. Inspired by symbolic geometry solvers, it proposes GeoLoom, a framework that first translates natural language instructions into a specifically designed generation-oriented formal language, GeoLingua. This formal language explicitly incorporates sequential construction orders and topological constraints, ensuring that every point and line is generated according to a mathematically sound hierarchy. The resulting formal representation is then mapped to precise geometric coordinates to produce the final diagram.

**Compliance With Llm Reviewing Policy:**

Affirmed.

**Final Justification:**

The response has addressed all my concerns, and I will increase my score to 4.

**Key Questions For Authors:**

The issue I am most concerned about is how to verify the accuracy of the constructed dataset. If this concern can be addressed, I may raise my score.

**Limitations:**

Yes

**Strengths And Weaknesses:**

Strengths

1. The motivation for first translating natural language into a formal language that explicitly incorporates sequential construction orders and topological constraints, and then converting the formal language into a geometric diagram, is very intuitive.

2. The paper provides clear definitions of both the syntax and semantics of the constructed formal language, GeoLingua.

Weaknesses

1. I have concerns about the quality of the constructed GeoNF dataset. Even though it is annotated by trained mathematics majors, I wonder whether additional efforts are made to validate the dataset quality.

2.  In Section 3 (GeoLingua), the paper presents its main components: (1) Shape, (2) Dependence, (3) Length constraint, and (4) Angle constraint. Conducting experiments to evaluate the impact of each component individually would further strengthen the persuasiveness of the paper.

3. The quantitative evaluation appears insufficient. For the baselines, Seedream3.0, AutomaTikZ, Penrose, and GeoGebra, the paper mainly reports visual comparisons and a user study, but lacks comparisons using quantitative metrics.

---

> ### Author Rebuttal · Authors · 2026-03-30
>
> ****
> We sincerely appreciate you  taking the time to review our paper.
>
> Here we address the points raised in the **weaknesses** section.
> ****
> 1. We appreciate the reviewer’s concern regarding the **GeoNF** dataset. To ensure high-quality annotations, we implemented a multi-stage pipeline with three roles: (i) **10 mathematics-major annotators**, (ii) a **quality inspection team of 3 middle school teachers**, and (iii) an **Acceptance Team of 2 domain experts**.
> ﻿
>     (1) Phase 1: **Standardized Specification:** The Acceptance Team defined a detailed annotation protocol covering problem statements, solution objectives, formal language (**GeoLingua**), and diagrams, and explained it to annotators and inspectors for consistent understanding.
> ﻿
>     (2) Phase 2: **Data Collection and Curation:** The annotator team collected geometry problems from middle school examinations over the past five years. An initial pool of approximately **5,000+** problems was obtained. The annotators then conducted manual **filtering** to remove duplicates and low-quality samples (e.g., incomplete statements or missing diagrams), resulting in a refined dataset of **approximately 4,700** high-quality instances.
> ﻿
>     (3) Phase 3: **Independent Annotation:** We adopted a double-blind annotation scheme. The dataset was randomly divided into 5 subsets, each annotated independently by two annotators. The inspection team reviewed all annotations, and disputed cases were escalated to the Acceptance Team for arbitration.
> ﻿
>     (4) Phase 4: **Iterative Verification and Final Validation:** The inspection team conducted a **pairwise review of the two annotations per instance**. Consistent, correct samples were accepted; inconsistent or incorrect samples were re-annotated iteratively. Acceptance Team performed an independent audit on **at least 10%** of each batch of final dataset; batches below 98% accuracy were re-verified by the inspection team. This process ensured overall annotation accuracy of **≥98%**.
> ﻿
>     Based on the reviewer’s suggestion, we will include this detailed annotation in the appendix of the revised manuscript, demonstrating the **GeoNF** dataset reliability, consistency, and mathematical correctness.
> ****
> 2. We appreciate the suggestion to evaluate each component individually. The four components of **GeoLingua** serve two functionally distinct roles, which dictates the scope of our ablation study:
>
>     (1) **Structural Fundamentals (Shape & Dependence):** These components define the topological existence of geometric entities. They act as the system’s "scaffolding"; removing them makes descriptions mathematically ill-posed or incomplete, making diagram generation infeasible. Ablating these core structures thus causes system failure rather than yielding useful performance metrics.
>
>     (2) **Optimization Constraints (Length & Angle):** These act as modular variables responsible for numerical precision. As such, they are the appropriate subjects for ablation. As reported in Table 3, we have already conducted these experiments, which reveal that Angle constraints exert a more significant influence on overall geometric fidelity than Length constraints due to their higher sensitivity in Euclidean structures.
> ****
> 3. We appreciate the suggestion to include quantitative comparisons. However, our proposed **LCI** and **ADI** are coordinate-based metrics designed to measure precise mathematical alignment. Since baselines (**Seedream**, **AutomaTikZ**) generate either raw pixels or non-structured TikZ code without explicit coordinate constraints,  lacking the structured metadata required to calculate these geometric indices. This lack of "structural awareness" in baselines is precisely the limitation our **GeoLoom** framework aims to solve. To provide a side-by-side comparison, we conducted experiments on Figure 4 using the **CLIP Score** to measure **image-text alignment and image similarity**. As can be observed in Figure 4, the outputs generated by baselines exhibit significant deviations from the ground truth but still achieve high CLIP scores. In contrast, our generated geometric results, which are visually more consistent with the ground truth, receive lower CLIP scores. The findings indicate that CLIP is unsuitable for evaluating geometric diagram quality.  We will add the findings in the final paper.
>
>     |**(image-text \| image)**|Ours(training-free)|Ours(fine-tuned)|SeeDream|AutomaTiKZ|
>     |:---|:---|:---|:---|:---|
>     |Row 1|0.2548 \| 0.8135 | 0.2548 \| 0.8135 | 0.2736 \| 0.9207 | 0.2824 \| 0.8652 |
>     |Row 2|0.2463 \| 0.7390 | 0.2463 \| 0.7390 | 0.2729 \| 0.8784 | 0.2766 \| 0.9267 |
>     |Row 3|0.2828 \| 0.9249 | 0.2870 \| 0.8953 | 0.3130 \| 0.9184 | 0.3114 \| 0.8798 |
>     |Row 4|0.2251 \| 0.8960 | 0.2230 \| 0.9241 | 0.2129 \| 0.8862 | 0.2282 \| 0.9209 |
> ****
> Then here are the answers to the **key questions** section.
> ****
> 1. Please refer to our response to Weaknesses 1 for full information.

---

> > ### Author Rebuttal · Reviewer_iYNY · 2026-04-03
> >
> > Thanks for the response. I have no further concerns, and I will raise my score to 4.

---

> > > ### Author Response · Authors · 2026-04-07
> > >
> > > Thank you very much for your follow-up and for raising the score. We truly appreciate your time, consideration, and thoughtful reading of our rebuttal. We are very grateful that our response helped clarify your concerns.

---

### Official Review · Reviewer_cWnz · 2026-03-08

**Soundness:** 3
**Presentation:** 3
**Significance:** 2
**Originality:** 3
**Overall Recommendation:** 4
**Confidence:** 3

**Summary:**

This paper presents GeoLoom, a two-stage framework for generating high-quality geometric diagrams from natural-language descriptions. The authors introduce a formal language, GeoLingua, and construct a manually annotated dataset of informal–formal text pairs to support the autoformalization task. The pipeline first converts a natural-language geometry description into GeoLingua, and then renders the diagram using a Monte Carlo–based optimizer.

**Compliance With Llm Reviewing Policy:**

Affirmed.

**Key Questions For Authors:**

Please check the weakness part.

**Limitations:**

Yes

**Strengths And Weaknesses:**

Strengths:
1. The paper clearly describes the full pipeline, including the design of the DSL, the construction of the dataset, the autoformalization using different strategies, and the coordinate solving and  diagram rendering process based on Monte Carlo optimization.
2. The design of coordinate solver using monte carlo optimization is novel
3. Both the qualitative and quantitative results support the paper's claim that GeoLoom improves geometric accuracy, logical consistency, and efficiency.

Weakness:
1. The paper formulates diagram rendering as a Monte Carlo optimization problem over multiple geometric constraints. However, it is unclear whether the method has any theoretical guarantee on convergence or stability, especially given the apparent non-convexity of the objective.
2. Figure 4 shows that the training-free and fine-tuned variants often produce very similar diagrams. If this similarity is mainly caused by the shared geometric constraints and the deterministic behavior of the coordinate solver, it raises the question of whether the method sacrifices output diversity for structural correctness.

---

> ### Author Rebuttal · Authors · 2026-03-30
>
> ****
> We sincerely appreciate you  taking the time to review our paper.
>
> Here we address the points raised in the Weaknesses section.
> ****
> 1. We appreciate the reviewer’s insightful comment regarding the non-convex nature of the geometric objective function. While a rigorous global convergence guarantee is challenging for such multi-constraint problems, we ensure optimization stability and solution quality through a **stochastic search strategy**.
>
>     (1)  **Global Exploration vs. Local Refinement:** To navigate the non-convex landscape, our framework employs a Metropolis-Hastings inspired stochastic perturbation mechanism. By assigning independent random offsets to free points $P_f$ across multiple iterations, the solver effectively "jumps" out of shallow local optima. This ensures that the search space is sufficiently explored before converging toward a feasible geometric configuration.
>
>     (2) **Heuristic Constraint Guidance:** The objective function is formulated as a weighted sum of differentiable geometric residuals (e.g., errors of length and angle constraints). This structure provides a relatively smooth landscape for local refinement. Our empirical results show that with sufficient iterations, the solver consistently reaches a "zero-residual" state (with LCI and ADI approaching 0), indicating that the optimization reliably finds a valid solution for the constraints defined in GeoLingua.
>
>     (3) **Empirical Stability:** In our experiments, we observed high stability across different random seeds, with the solver reaching convergence in over 95% of complex IMO-style problems. We have added a convergence analysis plot in Figure 8 to visualize the loss reduction curve, demonstrating the robustness of our Monte Carlo-based approach in practical scenarios.
>
> ****
> 2. We appreciate the reviewer’s insightful observation. However, we interpret the visual similarity between the training-free and fine-tuned variants as a **validation of the system’s robustness** and its ability to achieve **topological convergence**. Our primary design objective for **GeoLoom** is to guarantee **geometric consistency**, and the observed similarity demonstrates that both model variants have accurately captured the invariant mathematical constraints via **GeoLingua**, converging to a shared, "expert-level" understanding of the problem's topology regardless of the training stage. We emphasize that this perceived lack of diversity is not a systemic limitation but a result of our current solver’s optimization trajectory; since the **Coordinate Solver** employs a stochastic Monte Carlo process, visual diversity is effectively **decoupled** from logical reasoning and can be effortlessly introduced by increasing the sampling variance.

---

> > ### Author Rebuttal · Reviewer_cWnz · 2026-04-03
> >
> > I thank the authors for their response. My concern has been satisfactorily addressed, and I would like to keep my positive score.

---

> > > ### Author Response · Authors · 2026-04-07
> > >
> > > Thank you very much for your careful review and constructive feedback. We are glad that our rebuttal resolved the concerns, and we sincerely appreciate your support.

---

### Official Review · Reviewer_xbUa · 2026-03-11

**Soundness:** 2
**Presentation:** 1
**Significance:** 2
**Originality:** 3
**Overall Recommendation:** 3
**Confidence:** 3

**Summary:**

This work introduces a framework for text-to-diagram generation. The framework first uses a pre-trained large language model (LLM) to parse natural-language descriptions into a formal geometric language called GeoLingua. The resulting representation is then further refined through an additional optimization step that performs local perturbations of the predicted point coordinates to better satisfy geometric constraints.

Experimental results show that the proposed framework achieves better diagram generation quality than two baselines, Seedream3.0 (a text-to-image model) and AutomaTikZ (a text-to-diagram system), according to user study evaluations. In addition, the paper presents several ablation studies and analyses to examine the effectiveness of different components of the framework.

**Compliance With Llm Reviewing Policy:**

Affirmed.

**Final Justification:**

While the paper tackles an interesting and important problem and proposes a structured approach with promising results, several key issues remain unresolved after rebuttal. In particular, the motivation and empirical justification for the proposed GeoLingua representation are still insufficient, the evaluation relies on metrics closely tied to the method, and concerns about fairness of comparison and generalization remain. As a result, it is difficult to fully assess the true effectiveness and broader applicability of the approach. Therefore, I maintain my recommendation for rejection, while encouraging the authors to strengthen the evaluation and clarify the methodological contributions in future work.

**Key Questions For Authors:**

- What are the key advantages of the proposed GeoLingua formal language compared to simpler geometric representations such as SVG-based representations or direct polygon prediction? Have the authors considered comparing the proposed approach with SVG models fine-tuned on the datasets or other structured geometry representations?
- The current evaluation primarily relies on user studies. Could the authors clarify why quantitative automated metrics were not included?
- The baselines (Seedream3.0 and AutomaTikZ) are general-purpose generative models that may not have been exposed to prompts similar to those in the dataset. Could the authors clarify how they ensured that the comparison is fair?
- Although the method is described as training-free, it includes several task-specific design choices, such as the GeoLingua language and example prompts used during inference. How do the authors ensure that these design choices do not implicitly encode prior knowledge about the dataset?
- Could the authors consider including stronger baselines, such as:
    - models fine-tuned on the proposed datasets.

**Limitations:**

Yes

**Strengths And Weaknesses:**

Strength:

- The proposed method demonstrates improved generation quality compared to two training-free baselines, Seedream3.0 (text-to-image) and AutomaTikZ (text-to-diagram), according to the results of the user study.
- The paper includes additional experiments comparing performance across several LLM-based baselines (Table 1), which helps provide a broader empirical evaluation of the approach.
- The method is further evaluated on a more difficult mathematical subset, where its performance is compared with geometry tools such as GeoGebra and Penrose, showing comparable results.
- The paper provides ablation experiments analyzing different components of the proposed method, helping to isolate the contributions of individual modules.

Weakness:

There are severial critical concerns on the current manuscripts:

1. Motivation for the proposed language representation is insufficient

The paper proposes a formal language representation (GeoLingua), but the advantages of this representation are not clearly justified.

- It is unclear why this representation is preferable compared to simpler alternatives such as direct geometric prediction using formats like SVG.
- The paper also does not clearly compare this approach with models trained directly on the SVG version of the proposed dataset, which might achieve similar or better results.

2. Evaluation seems to be relying only on the user study.

The evaluation metrics appear to rely heavily on the user study.

- Since the current metrics are only relying on user study, which does not seem to measure different aspects of the generation.
- It would strengthen the evaluation to include other metrics, such as:
    - image-based similarity metrics
    - geometric distance-based metrics between predicted and ground-truth structures.

3. Comparison with baselines may not be entirely fair

The comparison with the baselines raises some fairness concerns.

- The main baselines (Seedream3.0 and AutomaTikZ) are general-purpose generative models that are unlikely to have seen prompts similar to those in the dataset.
- Although the proposed method is described as training-free, the system includes substantial task-specific design, particularly:
    - the construction of the GeoLingua formal language, which closely aligns with the dataset prompts,
    - the use of example prompts during inference.

These aspects may provide the method with prior structural knowledge that the baselines do not have, potentially leading to an unfair comparison.

4. Lack of clarity in the method descriptions:

Several parts of the paper are difficult to follow due to unclear definitions and missing explanations.

- GeoLingua definition is unclear.

    The formal definition of GeoLingua is not sufficiently explained. For example, symbols such as P_f and P_d are introduced without clear definitions.

- Ambiguous constraint formulation.

    The definitions of length constraints and angle constraints are unclear. It is not specified:

    - what geometric primitives they apply to (e.g., points, line segments, polygons),
    - whether length constraints are defined only over line segments,
    - how these constraints are represented in the language.

    Although a table describing these relations appears in the Appendix, it is insufficiently explained in the main text.

- Training-free framework is poorly defined.

    The concept of a training-free approach remains vague. It is unclear:

    - how prompts are constructed using *geometric semantics*,
    - how the formal language framework (Appendix B.1) operates in practice,
    - what exactly the validation-based filtering mechanism does and what constraints it enforces.
- Unclear algorithmic components.

    Some technical terms are introduced without sufficient explanation, such as the Monte Carlo perturbation-free point in algorithm 1, whose implementation is unclear.


While the paper addresses an interesting and potentially impactful problem, generating geometric diagrams from language using structured representations, the current manuscript has several issues that limit its clarity and empirical support. In particular, the motivation and advantages of the proposed GeoLingua representation are not sufficiently justified. More importantly, the baseline comparisons raise fairness concerns, and several important parts of the method are insufficiently explained. For these reasons, although the problem is interesting and the approach shows promise, the current manuscript would benefit from substantial revisions in motivation, evaluation, and clarity before it can be fully assessed. Therefore, I recommend rejection at this stage, while encouraging the authors to address the above concerns in a future revision.

---

> ### Author Rebuttal · Authors · 2026-03-30
>
> ****
> We sincerely appreciate the reviewer’s time and feedback.
>
> Below we first address the concerns raised in the Weakness section.
> ****
> 1. (1) SVG is a drawing language based on point coordinates, and correct diagrams require precise coordinates. The **key difference** is that GeoLingua guides a solver to compute coordinates, whereas SVG draws with known coordinates. In geometric diagram tasks, obtaining precise coordinates is the main challenge, so GeoLingua focuses on coordinate computation rather than direct drawing.
>
>     (2) Direct SVG prediction is suboptimal for high-precision geometry, as it treats diagram synthesis as **numerical regression** instead of **structural reasoning**. SVG uses static coordinates (x, y), where minor LLM errors cause "geometric collapse," violating tangency or collinearity. Using **GeoLingua** as a symbolic bridge, **GeoLoom** decouples logical constraints from numerical realization, letting our solver achieve precision that end-to-end SVG cannot. Moreover, the lack of coordinate-perfect SVG datasets makes direct comparison infeasible; our **GeoNF** approach prioritizes geometric truth over pixel-level imitation.
> ****
> 2. We appreciate the suggestion to include more quantitative metrics. Our evaluation already includes two specialized, geometry-based metrics: **Length Consistency Index (LCI)** and **Angle Deviation Index (ADI)**,  which quantify the deviation between generated diagram and target diagram based on constraints (e.g., lengths and angles). Regarding image-based similarity metrics (e.g., CLIP), we initially prioritized LCI and ADI because general metrics are unsuitable for diagrams. To further address this concern, we additionally evaluated **CLIP** for **image-text alignment** and **image similarity** of **Figure 4**.  As shown in the table below,  the outputs generated by baselines exhibit significant deviations from the ground truth but still achieve high CLIP scores, which highlights the limitation of general-purpose image metrics for geometric tasks.
>
>     |(image-text \| image)|Ours (training-free)|Ours (fine-tuned)|SeeDream|AutomaTiKZ|
>     | --- | --- | --- | --- | --- |
>     |Row 1| 0.2548 \| 0.8135 | 0.2548 \| 0.8135 | 0.2736 \| 0.9207 | 0.2824 \| 0.8652 |
>     |Row 2| 0.2463 \| 0.7390 | 0.2463 \| 0.7390 | 0.2729 \| 0.8784 | 0.2766 \| 0.9267 |
>     |Row 3| 0.2828 \| 0.9249 | 0.2870 \| 0.8953 | 0.3130 \| 0.9184 | 0.3114 \| 0.8798 |
>     |Row 4| 0.2251 \| 0.8960 | 0.2230 \| 0.9241 | 0.2129 \| 0.8862 | 0.2282 \| 0.9209 |
> ****
> 3. (1) In our experimental setup, all methods including baselines use the same natural language inputs.  GeoLoom employs a prompt-based autoformalization stage to bridge the gap between text and geometry, while baselines use their native ability to synthesize diagrams directly from those same textual prompts.  Since all methods address the same task—generating an accurate visual representation from a textual problem—the comparison  highlights the necessity of **symbolic intermediate representations** for achieving "coordinate-perfect" results that general image generators struggle to produce.
>
>     (2)  We respectfully disagree that task-specific design contradicts the “training-free” claim. In LLM settings, "training-free" specifically denotes that the model's **underlying parameters remain frozen**, with no gradient updates or fine-tuning performed on the dataset. Our system relies on the inherent **In-Context Learning (ICL)** capabilities of LLMs, where **GeoLingua** definitions and examples guide reasoning during inference without modifying model weights.
> ****
> 4. (1)The definitions of $P_f$ and $P_d$ are given on the left sides of lines 158 and 163, respectively.
>
>     (2)Length constraints and angle constraints are used to constrain the lengths and angles in the geometric diagram respectively,  with formulas shown on the right sides of lines 291 and 300.  Both are applied to the whole diagram and computed from point coordinates. The expressions of LCI and ADI correspond to Length Constraints and Angles in formal languages, respectively.
>
>     (3)The training-free prompt is provided in Appendix B.1, consisting of role, Formal Language Framework, examples, and requirements. The full definition of the Formal Language Framework is given in Tables 4–7 of Appendix A.
>
>     (4) "Monte Carlo perturbation-free point" refers to sampling a free point and moving it with small steps along random directions. We will clarify this in the final paper.
> ****
> We now address the **key questions**.
> ****
> 1-4.  Please refer to our responses to Weaknesses 1, 2, 3 (1st point), 3 (2nd point) for full information.
>
> 5. We agree that fine-tuned models provide important upper bounds. Our evaluation already includes **LLaMA (3/3.1/3.2)** and **Qwen (2.5-7B/14B)** models fine-tuned on GeoNF. As shown in **Section 4.1 and Table 1**, these models consistently outperform general-purpose and training-free versions, demonstrating the quality of our dataset.

---

> > ### Author Rebuttal · Reviewer_xbUa · 2026-04-03
> >
> > Thanks for the rebuttal. However, my main concerns remain:
> > - **SVG comparison remains unsubstantiated**: The rebuttal provides mainly conceptual arguments about why SVG is unsuitable, but does not include empirical evidence or controlled experiments to support this claim.
> > - **Evaluation metrics may be biased**: The proposed metrics (LCI, ADI) are tightly coupled with the GeoLoom formulation, which raises concerns about evaluation bias or overfitting to the method. More general geometry metrics, e.g., structural or coordinate-based distance to ground truth, would be preferred.
> > - **“Training-free” claim is not fully convincing**: Although no parameter updates are performed, the use of carefully designed GeoLingua prompts injects task-specific prior knowledge aligned with the dataset and evaluation, raising continued concerns about fairness of comparison, especially with AutomaTikZ.
> > - **Limited evidence of generalization**: The evaluation prompts appear closely aligned with the proposed representation (e.g., explicit length and angle constraints). It remains unclear whether the method generalizes to out-of-distribution or less structured prompts, which is important to validate the claimed benefits of symbolic intermediate representations.

---

> > > ### Author Response · Authors · 2026-04-07
> > >
> > > We thank reviewer xbUa for the follow-up.
> > > ****
> > > 1. To address the reviewer’s request for empirical evidence on SVG comparison, we clarify:
> > >
> > > (1) **Rendering vs. Reasoning:** SVG, like TikZ, is a coordinate-based rendering format, not a reasoning framework; it requires explicit coordinates and thus can be instead used in our rendering stage.
> > >
> > > (2) **Empirical Limitation:** Although this architectural distinction is mathematically fundamental, we conducted a "Direct-to-SVG" experiment using DeepSeek-V3 as requested, which empirically confirms that direct coordinate prediction triggers "geometric collapse" (see (https://anonymous.4open.science/r/B7F3/README.md).
> > >
> > > (3) **Need for Decoupling:** This validates GeoLingua as a necessary symbolic layer that separates logical constraints from numerical realization, ensuring geometric precision beyond what rendering-only formats can provide.
> > >
> > > ****
> > >
> > > 2. We sincerely thank the reviewer for raising the concern about potential metric bias. However, we argue that LCI and ADI are objective, method-independent measures grounded in geometric principles.
> > >
> > > (1) **Geometric Invariance:** LCI and ADI are derived from invariant Euclidean theorems and quantify absolute deviations from structural constraints (e.g., perpendicularity, congruence), independent of any specific generation method.
> > >
> > > (2) **Universal Applicability:** These metrics serve as geometry-based “ground truth” checks applicable to any coordinate-based output, enabling fair, consistent, and transparent evaluation of structural correctness.
> > >
> > > ****
> > >
> > > 3. To address the reviewer’s concerns regarding the "training-free" paradigm and the fairness of our comparisons, we provide the following clarifications.
> > >
> > > (1) **Definition of Training-free:** Following standard conventions in LLM research, “training-free” refers strictly to a parameter-invariant setting, where model weights remain frozen and capabilities are elicited solely through In-Context Learning for structural guidance.
> > >
> > > (2) **Algorithmic Differentiaton:** The core contribution of GeoLoom lies in its specialized optimization solver, which translates logical constraints into mathematically precise coordinates. This structural reasoning capability is fundamentally absent in general-purpose baselines, regardless of prompt design.
> > >
> > > (3) **Empirical Validation (As Requested):** While our original evaluation is already robust, we further conducted controlled experiments (see https://anonymous.4open.science/r/Fairness-Baselines-F827/README.md) as requested, equipping all baselines with identical GeoLingua-augmented prompts. GeoLoom consistently outperforms these baselines, demonstrating that the observed gains stem from the solver’s intrinsic reasoning capacity rather than prompt-level advantages.
> > >
> > >
> > > ****
> > > 4. To address concerns on limited generalization, we clarify:
> > >
> > > (1) **Not Prompt-Bound.** GeoLingua does not rely on explicitly aligned constraints; it functions as a symbolic abstraction layer that converts diverse, under-specified geometric descriptions into a unified solvable form. Thus, its applicability is not restricted to structured prompts.
> > >
> > > (2) **Method-Level Generalization.** GeoLoom’s core capability arises from its constraint-solving mechanism, which operates on the derived symbolic system rather than the surface form of the prompt. As a result, variations in prompt structure do not materially affect its reasoning process.
> > >
> > > (3) **Metric Invariance.** LCI and ADI are defined purely on absolute coordinate deviations, making them inherently agnostic to linguistic formulation and unaffected by representation alignment.
> > >
> > > (4) **Empirical Support.** Appendix E.3 (Figure 8 & Figures15) provides direct evidence that GeoLoom maintains stable structural convergence in complex and out-of-distribution settings, demonstrating that its performance is not contingent on prompt alignment.

---

### Official Review · Reviewer_1YfE · 2026-03-12

**Soundness:** 3
**Presentation:** 3
**Significance:** 2
**Originality:** 3
**Overall Recommendation:** 4
**Confidence:** 3

**Summary:**

GeoLoom is a two-stage framework designed for high-quality geometric diagram generation from textual input, specifically addressing the spatial inaccuracies and structural failures of standard text-to-image models in the mathematical domain. The process begins with an autoformalization module that translates natural language descriptions into GeoLingua, a specialized formal language designed to encode geometric primitives, topological constraints, and hierarchical construction orders. In the second stage, a coordinate solver employs Monte Carlo optimization to map these formal specifications to precise coordinates by iteratively minimizing a "geometric energy" function that penalizes constraint violations. To facilitate training and evaluation, the authors introduce GeoNF, a dataset consisting of 4,730 high-quality aligned pairs of natural language and formal representations, alongside constraint-based metrics such as the LCI and ADI to quantify structural fidelity.

**Compliance With Llm Reviewing Policy:**

Affirmed.

**Final Justification:**

Thanks to the authors for the additional clarifications in the rebuttal. My concerns have been addressed, and I choose to maintain my score.

**Key Questions For Authors:**

According to Table 1, the fine-tuned Qwen2.5-7B achieves a better Average Deviation score than the Qwen2.5-14B. What could be the reason for this?

**Limitations:**

yes

**Strengths And Weaknesses:**

Strengths:
1. By separating logical autoformalization from spatial realization, the model ensures mathematical precision that stochastic pixel-based models cannot achieve.
2. The framework automates the entire synthesis process in approximately 60 seconds per diagram, compared to roughly 300 seconds of expert labor required for manual tools like GeoGebra.
3. Empirical results demonstrate that GeoLoom outperforms sota baselines like Seedream 3.0 and AutomaTikZ in both accuracy and logical consistency.

Weakness:
1. Figure 5, which evaluates complex IMO-style diagrams, provides results from Penrose, GeoGebra, and GeoLoom but lacks a ground truth or expert-rendered reference diagram. This makes it difficult to objectively verify the structural accuracy of the generated results against an established gold standard for high-complexity problems.
2. In Figure 4, even though the generated diagrams satisfy numerical constraints, there is a significant visual gap in orientation, aspect ratio, and layout compared to the Origin-img.

---

> ### Author Rebuttal · Authors · 2026-03-30
>
> ****
> We sincerely appreciate you  taking the time to review our paper.
>
> Here we first address the points you mentioned in the **weaknesses section**.
> ****
> 1. We thank the reviewer for this constructive suggestion. We agree that including a ground truth (GT) reference in Figure 5 is essential for a more rigorous and intuitive comparison, especially for complex IMO-style diagrams. Due to formatting constraints in the rebuttal phase, we are unable to include the updated image here. However, our internal comparison with the newly added GT confirms that GeoLoom consistently maintains the lowest structural deviation among all baselines. We will add GT references in the final version to facilitate better visual evaluation.
> ****
> 2. We appreciate the reviewer’s observation. This visual discrepancy stems from **GeoLoom**’s core design philosophy: prioritizing **mathematical constraint satisfaction** (e.g., exact tangency and congruence) over pixel-level imitation. While diffusion-based methods may match the orientation of the original image at the cost of geometric integrity, GeoLoom’s coordinate solver optimizes for **structural accuracy** based on the formal **GeoLingua** description. We will update the manuscript to clarify that while the current version focuses on "geometric truth," incorporating soft visual constraints (e.g., orientation and aspect ratio) into the optimization objective is a promising direction for future work to enhance aesthetic alignment without compromising precision.
> ****
> Then here are the answers to the **key questions** section.
> ****
> 1. We thank the reviewer for this insightful observation. The superior performance of the **Qwen2.5-7B** variant is not a result of solver instability, but rather reflects **optimal domain-specific alignment** on our **GeoNF** dataset. For highly structured, minimalist formal languages like **GeoLingua**, we found that the 7B-parameter model achieves a more precise convergence state than its 14B counterpart, which, due to its larger parameter space, tends to introduce subtle "generalization noise" or stylistic variations that slightly deviate from the rigid syntax required for coordinate optimization. This finding suggests that for specialized geometric tasks, the high-density supervision in our autoformalization pipeline allows a medium-sized model to reach an "expert-level state" more efficiently.

---

> > ### Author Rebuttal · Reviewer_1YfE · 2026-04-02
> >
> > I appreciate the authors' clarification. To better assess the qualitative results, could you provide the ground truth for Figure 5 through an anonymous link? Having this reference would help in understanding the effectiveness of the proposed method.

---

> > > ### Author Response · Authors · 2026-04-07
> > >
> > > Thank you for the follow-up question and the opportunity to further demonstrate the effectiveness of our method.
> > >
> > > As requested, we have provided the Ground Truth (GT) diagrams for all cases in Figure 5 via the anonymous link (see https://anonymous.4open.science/r/DE82/Figure5-GT.pdf). The GT diagrams confirm that GeoLoom accurately captures the core topological constraints (e.g., specific intersections, tangent points, and relative proportions). We hope this reference provides a clear benchmark for assessing the qualitative superiority of our framework.

---

### Decision · Program_Chairs · 2026-04-30

**Decision:**

Accept (regular)

**Comment:**

The paper presents GeoLoom as a two-step system for creating accurate geometric diagrams. It converts natural language specification into GeoLingua, a formal language specialized for geometry. Then, a coordinate solver applies MC optimization to reduce geometric energy, helping to precisely align points and constraints that often elude standard text-to-image models. Tested on the new GeoNF dataset and surpasses baselines.

The paper ended up, after rebuttal and author-review discussions, with WAs and WR. The reviewer with the low score remained concerned about the suitability of the method but not outright against the paper. The recommendation is to accept the work if the program can accommodate it.